# Selective analysis of cancer-cell intrinsic transcriptional traits defines novel clinically relevant subtypes of colorectal cancer

Claudio Isella[1,2], Francesco Brundu[3], Sara E. Bellomo[1,2], Francesco Galimi[1,2], Eugenia Zanella[2], Roberta Porporato[2], Consalvo Petti[1,2], Alessandro Fiori[2], Francesca Orzan[1,2], Rebecca Senetta[2,4], Carla Boccaccio[1,2], Elisa Ficarra[3], Luigi Marchionni[5], Livio Trusolino[1,2], Enzo Medico[1,2,*] & Andrea Bertotti[1,2,6,*]

Stromal content heavily impacts the transcriptional classification of colorectal cancer (CRC), with clinical and biological implications. Lineage-dependent stromal transcriptional components could therefore dominate over more subtle expression traits inherent to cancer cells. Since in patient-derived xenografts (PDXs) stromal cells of the human tumour are substituted by murine counterparts, here we deploy human-specific expression profiling of CRC PDXs to assess cancer-cell intrinsic transcriptional features. Through this approach, we identify five CRC intrinsic subtypes (CRIS) endowed with distinctive molecular, functional and phenotypic peculiarities: (i) CRIS-A: mucinous, glycolytic, enriched for microsatellite instability or KRAS mutations; (ii) CRIS-B: TGF-β pathway activity, epithelial–mesenchymal transition, poor prognosis; (iii) CRIS-C: elevated EGFR signalling, sensitivity to EGFR inhibitors; (iv) CRIS-D: WNT activation, IGF2 gene overexpression and amplification; and (v) CRIS-E: Paneth cell-like phenotype, TP53 mutations. CRIS subtypes successfully categorize independent sets of primary and metastatic CRCs, with limited overlap on existing transcriptional classes and unprecedented predictive and prognostic performances.

[1] Department of Oncology, University of Torino School of Medicine, 10060 Candiolo Torino, Italy. [2] Candiolo Cancer Institute—FPO IRCCS, 10060 Candiolo Torino, Italy. [3] Department of Control and Computer Engineering, Torino School of Engineering, 10129 Torino, Italy. [4] Department of Medical Sciences, University of Torino School of Medicine, 10060 Candiolo Torino, Italy. [5] Department of Oncology, Johns Hopkins University, Baltimore, 21287 Maryland, USA. [6] National Institute of Biostructures and Biosystems, INBB, 00136 Rome, Italy. * These authors contributed equally to this work. Correspondence and requests for materials should be addressed to L.T. (email: livio.trusolino@unito.it) or to E.M. (email: enzo.medico@unito.it) or to A.B. (email: andrea.bertotti@unito.it).

A number of classification systems based on gene expression have been proposed that stratify colorectal cancer (CRC) in subgroups with distinct molecular and clinical features[1–7]. Comparative analyses in different data sets have revealed substantial classification coherence across the various signatures, particularly in the case of a 'Stem/Serrated/Mesenchymal' (SSM) subtype endowed with negative prognosis[8–10]. These classification efforts have been recently consolidated by a multi-institutional initiative that comprehensively cross compared the different subtype assignments on a common set of samples, leading to the definition of the consensus molecular subtypes[11] (CMS).

Interestingly, we and others independently reported that a large portion of the genes sustaining the SSM subtype (CMS4 within the CMS) are of stromal origin, and that the presence of stromal cells, mainly cancer-associated fibroblasts (CAFs), is a strong indicator of tumour aggressiveness[8,9]. Paradoxically, this could suggest that the non-neoplastic populations and the extrinsic factors of the tumour reactive stroma play the leading role in dictating cancer progression, while the intrinsic features of cancer cells convey less relevant cues. Alternatively, in whole tumour lysates the transcriptional consequences of biologically meaningful traits that are inherent to cancer cells might be obscured by the presence of a dominant, lineage-dependent transcriptional component of stromal origin. Indeed, an abundant tumour stromal content is expected to mask subtle gene expression profiles (GEPs) specifically exhibited by cancer cells. At present, very little is known about how and to what extent cancer cell-specific gene expression traits contribute to classify cancer. At least in principle, distilling variants based on genes that are expressed only by the transformed cells, in a context that is purified of heterologous multicellular complexity, might uncover subtypes that demonstrate higher predictive/prognostic value when used as classifiers.

To tackle this issue, we exploited a large collection ($n = 515$ samples from 244 patients) of patient-derived xenografts (PDXs)[12]. In PDXs, the stromal components of the original tumour are substituted by their murine counterparts as a consequence of xenotransplantation[13,14], so that detection of their transcripts can be avoided by appropriate use of human-specific arrays[8]. On these premises, here we leverage the unique opportunity afforded by PDXs to selectively explore the cancer cell-specific transcriptome of colorectal tumours. By doing so, we define the colorectal cancer intrinsic subtypes (CRIS) and evaluate their prognostic and predictive potential.

## Results

### CRC PDXs fail assignment to public transcriptional subtypes.

We and others have recently reported that CRC classification based on published transcriptional signatures is heavily affected by the tumour stromal content[8,9]. As a further evidence to this notion, we conducted principal component analyses on a large gene expression data set of primary colorectal tumours ($n = 450$)[15], assembled from information downloaded from the TCGA data portal (Supplementary Data 1), and confirmed that stromal genes have a major impact on the transcriptional profiles of such cases (Supplementary Fig. 1, Supplementary Data 2). To test whether this scenario extends beyond the context of primary tumours, we applied three published CRC signatures[1–3] to a proprietary data set ($n = 185$) of CRC liver metastases (CRC-LM) (GSE73255) using the nearest template prediction (NTP) algorithm[16]. This analysis showed that liver metastases behaved comparably to primary lesions, with a superimposable classification pattern (Fig. 1a, Supplementary Fig. 2a,b) that was sustained by similar transcriptional traits (Fig. 1b, Supplementary Fig. 2c,d). Overall, published signatures confidently classified

a large fraction of metastatic surgical specimens (range 80–95%, Fig. 1c, Supplementary Data 3).

We then analysed the GEPs of 515 PDXs from CRC-LM from 244 patients (GSE76402)[12]. To take into account intra-tumour heterogeneity, for most of the cases (149/244, 61%) multiple PDXs derived from regionally distinct areas of the original tumour were profiled (see Methods for details). A total of 115 of these patients (corresponding to 240 PDX profiles) were also included in the CRC-LM data set (Supplementary Data 4). In this setting, confident classification by published signatures was strongly reduced (range 50–90%; Fig. 1c, Supplementary Data 5). In particular, we observed a systematic loss of classification rate in the SSM classes, which had been previously reported to be mainly sustained by transcripts of stromal origin[8,9] ($\chi^2$-test assuming as null hypothesis that the fraction of cases classified as SSM in PDXs is not different from that obtained in the CRC-LM data set; $P < 5 \times 10^{-10}$, $\chi^2 = 41.084$, for CCS3; $P < 1 \times 10^{-4}$, $\chi^2 = 15.688$, for CRCA5; $P < 5 \times 10^{-09}$, $\chi^2 = 41.679$, for CCMS4). SSM classes also demonstrated the lowest validation rate when comparing the classification of CRC-LM with that of their corresponding PDXs (Fig. 1d,e, Supplementary Tables 3 and 5). We reasoned that this discrepancy in classification performance between original and mouse-propagated metastases might be ascribed to obliteration of human stroma in PDXs and substitution by host mouse components, which were not detected by human-specific arrays.

In line with previous genetic and phenotypic evidence[12,17], the classification incongruence between PDXs and their original counterparts is unlikely to be the consequence of genetic or functional drifts associated with PDX engraftment and propagation. Indeed, correlative analyses confirmed that the PDXs and the corresponding original CRC-LMs were significantly more similar than unmatched PDX/CRC-LM pairs (Wilcoxon rank sum $P < 5 \times 10^{-16}$, Fig. 1f). Moreover, in 88% of the cases, the best correlate for each PDX was its matched original counterpart (Supplementary Data 6). This suggests that PDXs largely maintain the transcriptional identity of their pre-implantation surgical pairs and puts forward the notion that depletion of stroma-derived signals is likely the major source of transcriptional variation between surgical specimens and PDXs.

### Cancer-cell intrinsic transcriptional traits classify CRC.

The tumour stroma certainly contributes to tumour biology but at the same time is not a direct expression of the transformed phenotype of neoplastic cells. Furthermore, classical histopathological studies have shown that many solid tumours (including CRC) feature a desmoplastic reaction whereby dense connective tissue, produced by activated fibroblasts, tends to prevail quantitatively over areas occupied by cancer cells[18,19]. We therefore speculated that stromal content might act as a dominant source of variation that could mask more subtle—albeit biologically relevant—traits inherently related to properties of cancer cells. To explore this hypothesis, we took advantage from PDX-derived GEPs to uncover cancer-cell intrinsic transcriptional subtypes in CRC.

By applying non-negative matrix factorization (NMF)[20] to the 515-PDX expression data set, we identified optimal partitioning into five clusters (Fig. 2a, Supplementary Data 7). As previously described, we employed significance analysis of microarrays (SAM[21]; FDR < 0.005) and prediction analysis for microarrays (PAM[22]; FDR = 0.035) to identify 903 genes whose expression best discriminates each subtype. To generate a cancer-cell intrinsic classifier that was not influenced by stroma-derived transcripts in human CRC samples, we excluded from the

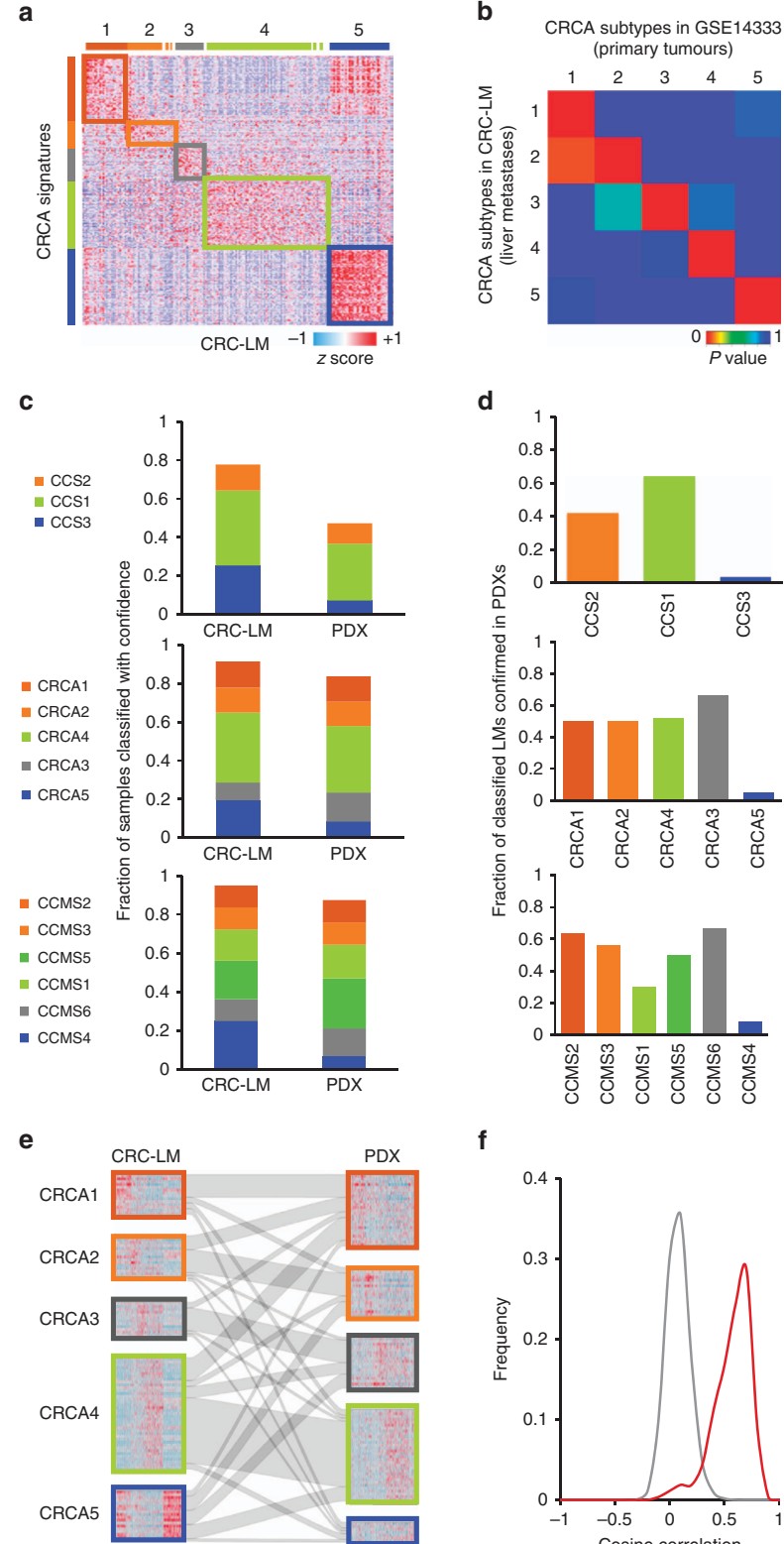

**Figure 1 | CRC PDXs fail assignment to public transcriptional subtypes.** (**a**) Heatmap showing NTP classification of the CRC-LM data set based on the signatures of the CRCA classification system. (**b**) Submap analysis of CRCA subtype similarity between CRC liver metastases (CRC-LM) and primary tumours (GSE14333). *P* values are calculated by Fisher's exact test using the Submap tool available from Gene Pattern. (**c**) Column chart showing the fractions of CRC liver metastases (CRC-LM), or their corresponding PDXs, which were confidently assigned (NTP, FDR < 0.2) to the subtypes of three different public classifiers. (**d**) Fraction of PDXs assigned to the same class of their corresponding liver metastasis according to the three classifiers (as in **c**). (**e**) Caleydo view of correspondences between the CRCA class assignments of CRC-LM samples and those of their PDX counterparts. (**f**) Distribution of Pearson's correlation values obtained by analysing unmatched (grey line) and matched (red line) CRC-LM/PDX pairs.

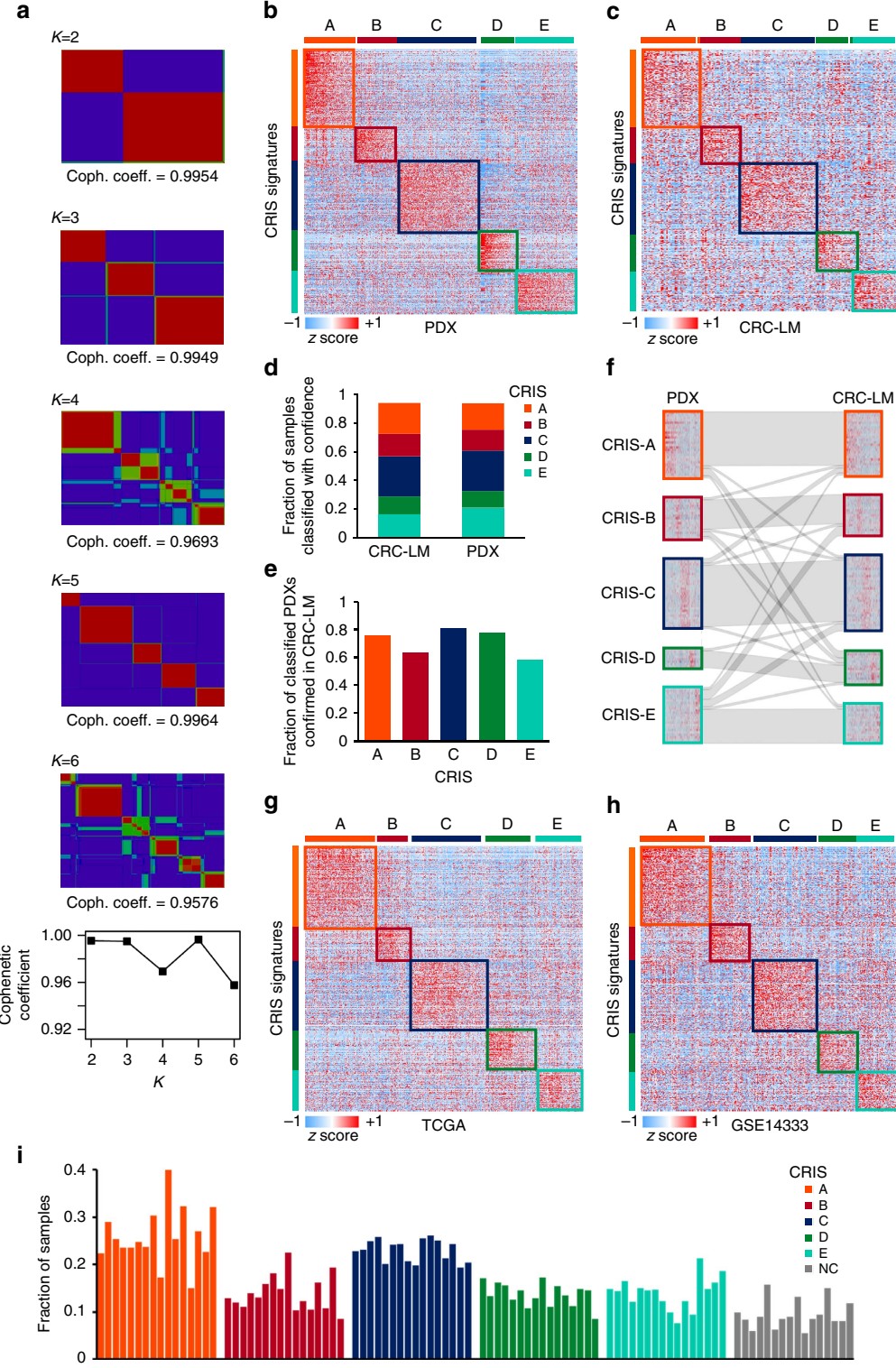

**Figure 2 | CRIS classifies both CRC tumours and PDXs based on cancer-cell intrinsic transcriptional traits.** (**a**) NMF-based consensus clustering of the PDX data set and plot of cophenetic coefficients for $K = 2$–6. (**b,c**) Heatmap showing NTP-based CRIS classification of the PDX (**b**) and the CRC-LM (**c**) data sets. (**d**) Column chart showing the fractions of CRC-LMs and PDXs that were confidently classified (FDR < 0.2) by CRIS; coloured sub-columns represent the relative distribution of subtypes within each data set. (**e**) Column chart showing, for each CRIS subtype, the fractions of samples for which CRC-LMs and their corresponding PDXs were confidently assigned (FDR < 0.2) to the same class. (**f**) Caleydo view of CRIS correspondences between CRC-LM samples and their PDX counterparts. (**g,h**) Heatmaps showing CRIS classification of primary tumours based on RNAseq data from the TCGA (**g**) and Affymetrix microarray data of the GSE14333 data set (**h**). (**i**) Column chart showing the distribution of CRIS subtypes based on NTP classification across 16 independent CRC data sets (Supplementary Data 11); NC (non-classified) indicates samples for which NTP could not confidently assign any of the subtypes (FDR > 0.2).

classifier all the transcripts for which 50% or more of the signal was originated from stromal cells, based on the analysis of PDX RNAseq data (Methods). By applying this filter, we selected 565 genes with unambiguous epithelial expression (Supplementary Data 8). When applied to the PDX training data set (Fig. 2b), CRIS confidently classified 94% of the samples (Supplementary Data 7). Importantly, similar classification results were obtained in the original CRC-LM data set (Fig. 2c, Supplementary Data 9). All CRIS subtypes maintained a similar fraction of assigned cases in the PDX and CRC-LM data sets (Fig. 2d). Submap analysis[23] confirmed that each subtype was associated with similar underlying transcriptional traits in the PDX and CRC-LM data sets (Supplementary Fig. 3). This indicates that the transcriptional patterns identified in PDXs are represented and detectable in their original counterparts, even in the presence of human stromal cells. This notion is further reinforced by the high level of concordance observed when comparing the classification of matched PDXs versus original tumour pairs (75% and kappa score = 0.62 for CRIS versus 28–46% and kappa scores ranging from 0.2 to 0.3 for existing classifiers) (Fig. 2e,f; compare with Fig. 1d,e).

Because CRIS was derived from metastatic samples grown as xenografts in mice, we wished to verify whether CRIS signatures could be also applied to CRC gene expression profiles obtained in different conditions and by means of diverse technologies. To this aim, we initially tested the CRIS classifier against two independent gene expression data sets of primary CRCs, the previously mentioned 450-sample TCGA RNAseq data set[8,15] (Supplementary Data 1) and GSE14333, a 290-sample Affymetrix data set[24] (Fig. 2g,h, Supplementary Data 10). In both instances, more than 90% of the samples were significantly assigned to CRIS classes (Supplementary Data 11). In the case of the TCGA data set, for which microsatellite instability status was available, CRIS classification efficiency was high in both microsatellite-stable (MSS) and microsatellite-unstable (MSI) samples (93% and 95% of confidently classified samples, respectively). This was particularly interesting because the CRIS training data set had very few MSI cases (Supplementary Data 4), as expected for metastatic CRC[25]. For additional independent CRIS validation, we applied the classifier to 14 other data sets[11,26], including one Affymetrix microarray data set of liver metastases and one Illumina microarray data set of CRC cell lines (Supplementary Data 11). Overall, CRIS confidently classified 3,396 out of 3,738 samples (90.8%). Further, the five subtypes maintained a similar fraction of assigned cases, irrespective of technological platforms and experimental conditions (Fig. 2i). Submap analysis confirmed that the transcriptional attributes distinguishing the different subgroups were comparable in CRC-LM and primary tumours (Supplementary Fig. 4). Collectively, these results indicate that the transcriptional traits captured by CRIS reflect stable intrinsic features of cancer cells and are not ostensibly affected by the origin of the samples or the expression analysis platform.

**Major peculiarities of CRIS classes.** Using NTP-calculated centroid distances in the TCGA data set (Supplementary Data 11), CRIS classes could be clustered into two major subfamilies composed of two (CRIS-A and CRIS-B) and three (CRIS-C, CRIS-D and CRIS-E) members (Fig. 3a; Methods). As suggested by the asymmetric distribution of mutational and copy number load (Fig. 3b,c), TCGA MSI and MSS samples (Supplementary Data 1) were not equally partitioned across CRIS subfamilies. Specifically, MSI tumours were predominantly assigned to CRIS-A and—to a lower extent—to CRIS-B (Fisher's exact test, CRIS-A/B against all other samples, $P < 5 \times 10^{-10}$,

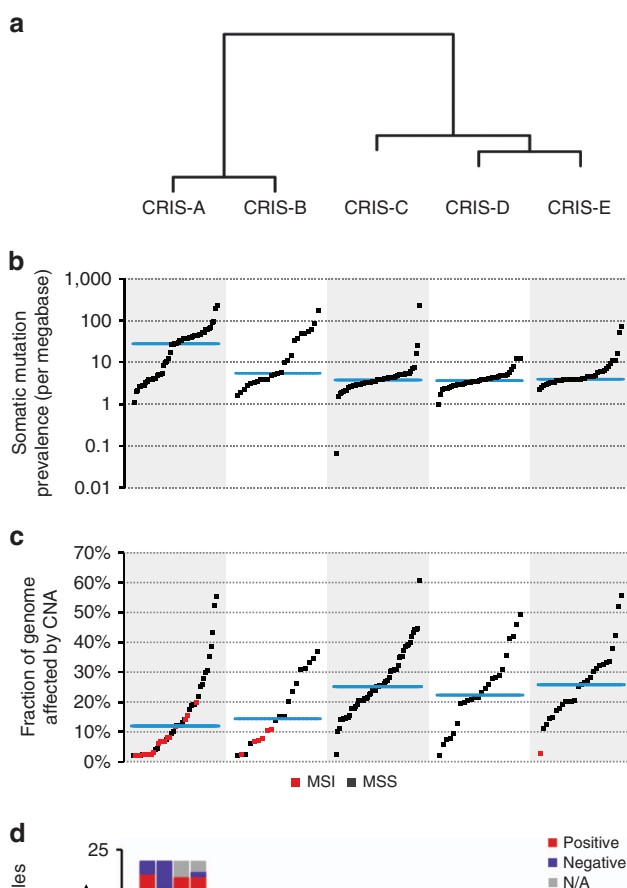

**Figure 3 | Molecular and phenotypic characteristics of CRIS subtypes in the TCGA data set.** (**a**) Clustering of CRIS classes based on similarity of the distances, as calculated by NTP, when classifying the TCGA data set. (**b**) Frequency of somatic single-nucleotide variations. Individual samples are represented as black dots; the mean of each CRIS subtype is plotted in light blue. (**c**) Extent of somatic copy number alterations. Samples are represented as red or black dots to denote MSI or MSS tumours, respectively; the mean of each CRIS subtype is plotted in light blue. (**d**) Distribution of the indicated phenotypic and genetic characteristics of CRC samples across CRIS subtypes.

odds ratio 17.309, confidence interval (CI) = 5.670–71.189; Fig. 3d). These two classes proved to be also enriched for right-colon tumours featuring mucinous histology, CpG island methylator phenotype (CIMP) and a hypermutator genotype (Fisher's exact test, CRIS-A/B against all other samples, $P < 5 \times 10^{-3}$, odds ratio = 2.701, CI = 1.459–5.038 for right-colon location; $P < 5 \times 10^{-10}$, odds ratio = 16.207, CI = 5.751–56.814 for mucinous histology; $P < 1 \times 10^{-11}$, odds ratio = 27.254, CI = 7.811–147.338 for CIMP; $P < 5 \times 10^{-7}$, odds ratio = 16.094, CI = 4.388–89.516 for hypermutator phenotype; Fig. 3d). Of note, some of these MSI-associated features were also shared by the MSS members of CRIS-A ('MSI-like' samples; Fisher's exact test, CRIS-A MSS samples against all other MSS samples, $P < 5 \times 10^{-6}$, odds ratio = 23.231, CI = 5.318–144.640 for mucinous histology and

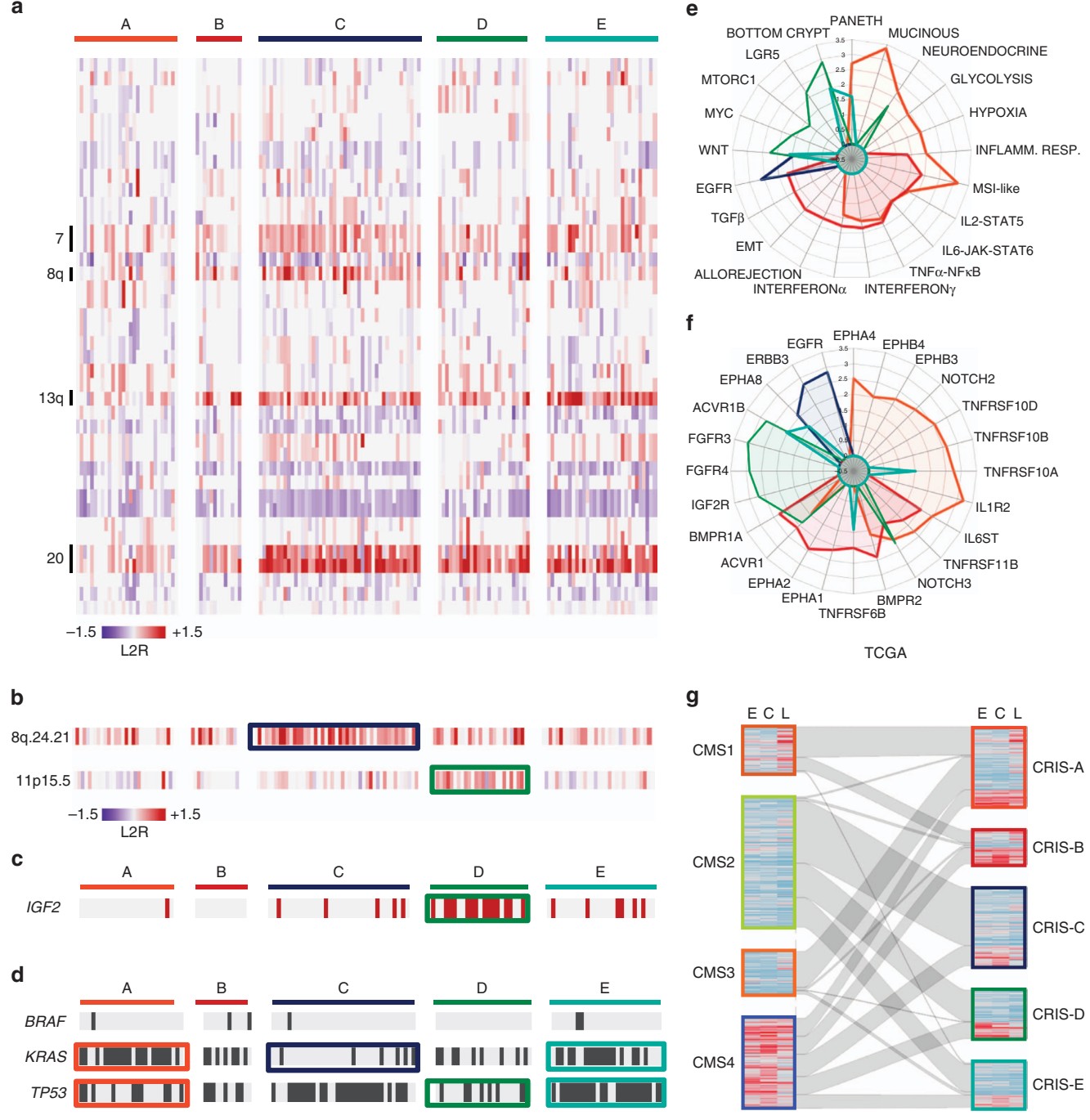

**Figure 4 | Genetic and functional features of CRIS classes in MSS colorectal cancer.** (**a**) Heatmap showing the distribution of broad copy number changes displaying the top-five highest average variations in individual CRIS subtypes in MSS samples of the TCGA data set. (**b**) Heatmap showing selected focal amplifications in MSS samples of the TCGA data set. Significant subtype enrichments are marked by coloured boxes. (**c**) Distribution of cases overexpressing *IGF2* according to the TCGA data set[15]. (**d**) Distribution of sequence alterations in *BRAF*, *KRAS* and *TP53* in MSS samples extracted from the TCGA data set. Significant subtype enrichments are marked by coloured boxes. For enrichment analysis, $P < 0.05$ by Fisher's exact test was considered significant. (**e**) Radar plot showing phenotypic and functional characteristics of CRIS subtypes, expressed as normalized enrichment scores (NESs) calculated by GSEA. (**f**) Radar plot showing autocrine stimulation loops in CRIS subtypes, expressed as NESs and calculated based on gene expression analysis of ligand-receptor pairs. (**g**) Caleydo view of correspondences between CMS subtypes and CRIS classes in the TCGA data set. Heatmaps show estimates of stromal infiltration derived from gene expression analysis of specific signatures (C, CAFs; E, endothelial cells; L, leukocytes).

$P < 5 \times 10^{-3}$, odds ratio $= 10.417$, CI $= 2.031$–69.611 for CIMP; Fig. 3d).

Through an exploratory inspection of broad and focal copy number changes specifically in MSS samples, we identified genomic traits peculiar to individual CRIS classes (Fig. 4a,b).

Overall, CRIS-C, D and E typically displayed chromosomal instability (CIN), characterized by heavy copy number alteration burden (Wilcoxon rank-sum test, $P < 5 \times 10^{-08}$; Fig. 3c). More specifically, gains of entire chromosome arms were particularly evident for CRIS-C (for example, Chr7, Chr8q and Chr20) and

CRIS-E (Chr13q) (Fig. 4a). Interestingly, CRIS C–D–E subtypes displayed significant association with specific focal amplifications (Supplementary Table 1). In particular, besides some alterations with unknown functional consequences, CRIS-C displayed focal amplification of 8q.24.21, which contains the *MYC* proto-oncogene (Fisher's exact test, CRIS-C samples against all other MSS samples, $P < 0.01$, odds ratio = 2.802, CI = 1.207–7.006; Fig. 4b). CRIS-D was specifically enriched for amplification of Chr11p15.5 (Fisher's exact test, CRIS-D samples against all other MSS samples, $P < 1 \times 10^{-05}$, odds ratio 6.020, CI = 2.268–17.046; Fig. 4b). This locus contains *IGF2*, a gene whose strong overexpression has been reported in a subset of CRCs[15]. Accordingly, we observed that Chr11p15.5 amplification in CRIS-D was paralleled by high *IGF2* expression levels (Fig. 4c).

To further investigate the genetic correlates of CRIS transcriptional classes, we took advantage of the mutational profiles available from the TCGA data portal[15]. As previously reported[27], the prevalence of mutations in the *BRAF* gene was high in MSI samples (Supplementary Data 1). While the few MSS cases with *BRAF* alterations did not cluster in any particular CRIS subtype (Fig. 4d), 13 of the 15 MSI samples with mutated *BRAF* were assigned to CRIS-A (one with high FDR) and two to CRIS-B. Within MSS samples, *KRAS* was frequently mutated in CRIS-A (Fig. 4d; Fisher's exact test, CRIS-A MSS against all other MSS samples, $P < 0.005$, odds ratio = 4.320, CI = 1.570–13.274) and, within the CRIS C–D–E subfamily, specifically in CRIS-E (Fisher's exact test, CRIS-E against CRIS-C/D, $P < 5 \times 10^{-4}$, odds ratio = 5.645, CI = 2.059–16.120; Fig. 4d). CRIS-C was instead strongly enriched in *KRAS* wild-type samples (Fisher's exact test, CRIS-C against all other MSS samples, $P < 1 \times 10^{-4}$, odds ratio = 0.161 CI = 0.050–0.441; Fig. 4d). Finally, *TP53* mutations prevalently occurred in CRIS-E (Fisher's exact test, CRIS-E against all other MSS samples, $P < 0.01$, odds ratio = 3.892, CI = 1.307–14.156; Fig. 4d), while they were depleted in CRIS-A and D (Fisher's exact test, CRIS-A or D against all other MSS samples, $P < 0.05$ for both, odds ratio = 0.361 with CI = 0.132–0.942 for CRIS-A and 0.281 with CI = 0.095–0.771 for CRIS-D; Fig. 4d). By leveraging the genetic annotations available in our data set of CRC PDXs (Supplementary Data 4), we were able to confirm several of the molecular correlates found in the TCGA cohort (Supplementary Fig. 5). Together, these findings indicate that CRIS subclasses are endowed with specific genetic traits that presumably drive functional features associated with the corresponding expression signatures.

To better delineate the functional attributes inherent to CRIS subtypes, we exploited both unbiased and supervised approaches. On the one hand, to gather a comprehensive picture of biological traits associated with CRIS classes, we ran gene set enrichment analysis (GSEA) against the 'Hallmark' gene sets, a panel of signatures representing an extensive spectrum of key biological functions, from the MSigDB (ref. 28) (Fig. 4e, Supplementary Data 12). On the other hand, to explore transcriptional patterns specifically relevant in CRC, we selected ten signatures capturing different phenotypic or functional aspects of normal and neoplastic intestinal biology (Supplementary Data 13) and analysed to what extent each sample displayed the phenotype captured by every signature. To do this, for each sample and each signature a transcriptional score was calculated by subtracting the average expression of the genes negatively associated with the phenotype from the average expression of the genes positively associated with the phenotype. Then, to test how the different CRIS subtypes are enriched for samples strongly displaying the selected transcriptional signature, we conceived a procedure named 'Sample Set Enrichment Analysis' (SSEA). In SSEA, samples are ranked by the calculated signature scores, then the

GSEA algorithm is employed to test 'sample sets' (in our case samples belonging to each CRIS subtype) for enrichment in high-rank samples (Fig. 4e, Supplementary Data 13; see Methods). Using SSEA, we also estimated the enrichment of each subtype in mitogenic/anti-apoptotic autocrine loops, as a proxy of growth factor-dependent oncogenic signalling. In this case, the samples were ranked based on 'receptor activity scores', which were calculated by averaging the expression of the receptor itself with that of its ligands (Fig. 4f, Supplementary Data 14; Methods). To avoid the confounding effect due to dilution of cancer-cell specific transcripts by stroma-derived RNA, all these analyses were performed on PDX GEPs.

Both GSEA-based and SSEA-based phenotypic analyses were concordant in attributing secretory and MSI-like features to CRIS-A, in agreement with the mucinous histology of this subgroup detected in the TCGA data set (Fig. 4e). CRIS-A was also associated with inflammatory traits and a glycolytic/hypoxic status (Fig. 4e), which is typical of some *KRAS*-mutated tumours[29] and has been linked to intense secretory activity[30,31]. Similar to CRIS-A, CRIS-B featured an inflammatory phenotype (Fig. 4e); but different from CRIS-A, CRIS-B also displayed strong TGFβ activity and marked traits of epithelial–mesenchymal transition (EMT) (Fig. 4e). This is consistent with the established role of TGFβ as an EMT inducer[32]. Interestingly, blinded pathological inspection of images from the TCGA cancer digital slide archive[33] revealed that CRIS-B members included a large number of poorly differentiated tumours, in which the glandular architecture of the original tissue was completely lost or barely detectable (Supplementary Fig. 6). These preliminary observations deserve future, more extensive exploration. CRIS-C was mainly characterized by a combination of elevated ERBB1/3 autocrine stimulation loops and moderate WNT (Fig. 4e,f). CRIS-D showed traits of high intrinsic IGFR stimulation (in accordance with the strong prevalence of *IGF2* amplification observed in the TCGA data set) and FGFR activity (Fig. 4f). Finally, CRIS-D—and, to a lesser extent, CRIS-E—featured high WNT activity and a bottom crypt phenotype. In particular, CRIS-D was positive for an LGR5 signature, which typifies intestinal stem cells[34], and CRIS-E displayed a phenotype recalling WNT-producing Paneth cells (Fig. 4e,f).

In summary, CRIS identified CRC subclasses with different functional and phenotypic characteristics related to specific molecular alterations: CRIS-A was an MSI-like, *BRAF*- or *KRAS*-mutated, and secretory subtype with sustained glycolytic metabolism and inflammatory traits; CRIS-B grouped a subset of poorly differentiated tumours with active TGFβ signalling and EMT features; CRIS-C clustered *KRAS* wild-type CIN tumours exhibiting elevated ERBB/EGFR pathway activity and *MYC* copy number gains; CRIS-D tumours were characterized by a typical stem phenotype with high WNT signalling, associated with strong enrichment in *IGF2* amplification/overexpression and FGFR autocrine stimulation; CRIS-E again displayed WNT-related features, but associated with a Paneth-like phenotype and a *TP53*-mutated genotype.

Of note, many of these characteristics were not reported to associate with transcriptional subgroups in previous studies[1–7], strongly suggesting that the elimination of stroma-related effects during the class discovery process enhanced sensitivity towards detection of intrinsic cancer cell-specific traits. To further support this notion, we analysed the distribution of CRIS subtypes across CMS classes assigned to TCGA samples[11]. Overall, partition of the samples by the two classifiers displayed limited overlap (Fig. 4g; see Supplementary Table 2 for cross-tabulation and statistics). Most CMS1 samples were assigned to CRIS-A and B; CMS2 samples were partitioned in CRIS-C, D and E; CMS3 contributed mostly to CRIS-A; and finally, CMS4 was

orthogonally distributed across all five CRIS classes, with an enrichment for CRIS-B cases. These differences are largely attributable to the influence of stromal infiltration, as evidenced by analysing the distribution of stromal signatures individually expressed by CAFs, leukocytes or endothelial cells[8] (C, L and E stromal scores, respectively) in the two classifiers. As shown by

the heatmaps in Fig. 4g, asymmetric expression of stromal signatures (presented as the average expression of the component genes of each signature) was clearly detected in the CMS partition but not in the CRIS partition. Indeed, CMS1 samples displayed a high L score, in line with the fact that *BRAF*-mutant MSI CRC tumours, mostly included in CMS1, typically exhibit an extensive immune cell infiltration[11] (Fig. 4g). Conversely, CRIS-A comprises *BRAF*-mutant MSI tumours but also *KRAS*-mutant MSS samples with MSI-like features, and overall has distinctive metabolic features. Accordingly, only a fraction of CRIS-A samples had a high L score, and this class received additional samples from CMS3, encompassing tumours with mixed MSI status, enrichment for *KRAS* mutations, and metabolic deregulation, but devoid of intense leukocyte infiltration. Most CMS4 tumours are characterized by very high levels of all stromal scores, in particular of the C score (Fig. 4g). The absence of overlaps with CRIS classes corroborates the notion that assignment to CMS4 mostly depends on stromal transcript contribution rather than intrinsic cancer-cell features. A fraction of the CMS1 and CMS4 samples converge into the CRIS-B group, featuring EMT and high TGF-β signalling; these pro-invasive traits are indeed assigned by CMS only to the CSM4 class, probably because they are estimated by stroma-derived transcripts. A similar lack of overlap with CRIS classification and asymmetric distribution of stromal scores across subtypes was also observed when previously published transcriptional CRC classifiers were applied to the TCGA data set (Supplementary Fig. 7).

The sensitivity of CRIS for more granular detection of functionally relevant cancer-cell intrinsic traits is particularly evident for CMS2. This subgroup, the vastest of the consensus, includes CIN tumours that show overall upregulation of WNT targets, and displays consistently low stromal scores. The CMS2 subtype was resolved by CRIS into three distinct groups that displayed class-specific features superimposed on a common background of high WNT signalling: CRIS-C is characterized by high EGFR pathway activity; CRIS-D is enriched for IGF2 overexpressors; CRIS-E contains a high number of samples with *KRAS* and *TP53* mutations. Collectively, these findings further attest to the independent value and higher resolution of CRIS taxonomy.

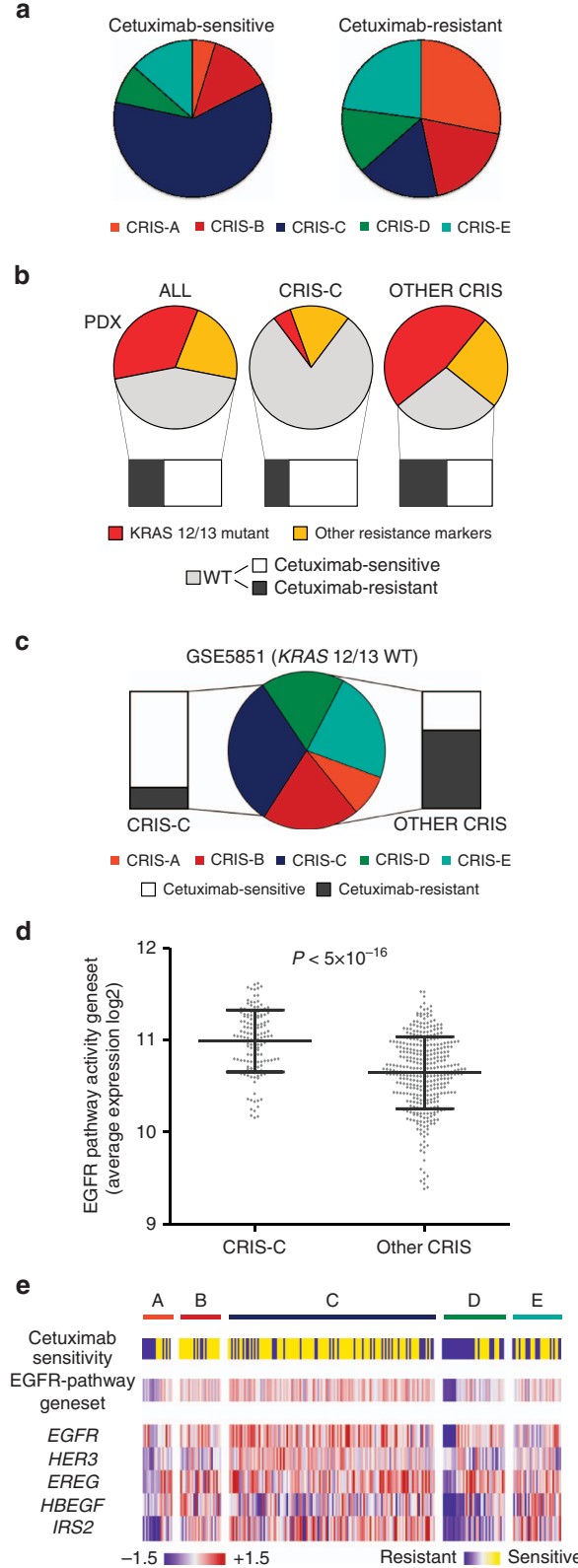

**Figure 5 | CRIS-C and EGFR pathway activity predict cetuximab sensitivity.** (**a**) Prevalence of CRIS classes among PDX cases scored as sensitive or resistant to cetuximab. (**b**) Bar-of-pie charts showing the distribution of validated resistance biomarkers (*KRAS* mutations in codons 12 and 13) and proposed resistance biomarkers (other resistance markers; for their enumeration, see Supplementary Data 4) in PDXs (pies). The fraction of sensitive or resistant cases specifically in the PDX subpopulation that does not harbour resistance biomarkers is shown in the bars. The analysis encompasses the whole PDX data set, CRIS-C alone, and all other CRIS classes. (**c**) Bar-of-pie chart showing the distribution of CRIS subtypes in *KRAS* wild-type cases of GSE5851, a data set of CRC liver metastases annotated for clinical response to cetuximab monotherapy[37] (pie). Bars denote the fraction of tumours responsive or resistant to cetuximab within CRIS-C or in all other CRIS classes. (**d**) Average expression of the gene set indicative of EGFR pathway activity in samples assigned to CRIS-C versus samples assigned to all other CRIS classes; error bars represent s.d. from the mean; statistical significance was calculated by Wilcoxon rank-sum test. (**e**) Heatmap showing response to cetuximab, average expression of the gene set indicative of EGFR pathway activity, and expression of individual genes of the EGFR pathway activity gene set across CRIS classes. The analysis includes PDXs that do not harbour known genetic markers of resistance.

**CRIS predicts response to anti-EGFR antibodies**. To test whether CRIS partitioning could predict drug sensitivity, we deployed expression profiles from 241 PDXs annotated for their response to the anti-EGFR antibody cetuximab[12,35,36] (Supplementary Data 4, Supplementary Fig. 8). Interestingly, CRIS-C was significantly over-represented among cetuximab-sensitive tumours and depleted from resistant cases (Fisher's exact test, $P < 5 \times 10^{-10}$, odds ratio = 8.23, CI = 4.017–17.545; Fig. 5a). The opposite was true for CRIS-A, which correlated with lack of response (Fisher's exact test, $P < 1 \times 10^{-5}$, odds ratio = 0.104, CI = 0.020–0.354; Fig. 5a). We and others have identified a number of genetic alterations that associate with resistance to cetuximab[12,36–39]. These alterations, particularly the clinically validated mutations in *KRAS* and *NRAS* genes, were under-represented in CRIS-C (Fisher's exact test for *KRAS* and *NRAS*, $P < 1 \times 10^{-13}$, odds ratio = 0.066, CI = 0.022–0.163; Fig. 5b). In principle, this could suggest that the higher rate of cetuximab-sensitive CRIS-C tumours is due to depletion in cases harbouring resistance-conferring alterations. However, CRIS-C maintained strong predictive power also when such cases were not factored in the analysis (Fisher's exact test, $P < 1 \times 10^{-5}$, odds ratio = 3.971, CI = 2.08–7.08; Fig. 5b). As an external validation, we tested CRIS classification on a published data set of CRC-LM annotated for clinical response to cetuximab[40] (GSE5851; Supplementary Data 15). Again, CRIS-C predicted cetuximab sensitivity both in the whole cohort (Fisher's exact test, $P < 0.005$, odds ratio = 6.780, CI = 1.809–29.62; Supplementary Fig. 9) and, notably, also when *KRAS*-mutated samples were excluded from the analysis (Fisher's exact test, $P < 0.05$, odds ratio = 6.61, CI = 1.07–74.094; Fig. 5c).

Complementary to genetic markers of resistance, the expression of a set of genes indicative of EGFR pathway activity has been found to correlate with cetuximab sensitivity[35,36,40]. This gene set displayed consistently higher expression in CRIS-C (Wilcoxon rank-sum test, CRIS-C samples against all other samples, $P < 1 \times 10^{-8}$; Fig. 5d,e; Supplementary Fig. 10; Supplementary Tables 4 and 9). Moreover, expression of the gene set was significantly associated with cetuximab sensitivity across the whole data set, even after exclusion of all genetic markers of resistance (Wilcoxon rank-sum test, $P < 1 \times 10^{-4}$; Supplementary Fig. 11a,b) and—of note—also when CRIS-C cases were excluded (Wilcoxon rank-sum test, $P < 5 \times 10^{-4}$; Fig. 5e; Supplementary Fig 11c). Multiple logistic regression analysis demonstrated that CRIS classes, genetic markers and EGFR pathway activity transcripts have independent predictive value (Supplementary Table 3). This suggests that, although partially overlapping, these layers of information convey integrative knowledge, which could be combined to obtain more accurate prediction of cetuximab sensitivity.

**CRIS is an independent predictor of CRC prognosis**. To explore CRIS prognostic impact we examined the association of CRIS subtypes with disease-free survival (DFS) of 290 patients included in a publicly available and clinically annotated CRC data set[24] (GSE14333; Supplementary Tables 10 and 11). The analysis revealed a trend towards better prognosis for CRIS-D tumours and a significant association with worse prognosis for CRIS-B tumours (log-rank $\chi^2$, $P < 5 \times 10^{-4}$, hazard ratio (HR) = 2.961, CI = 1.615–5.43; Fig. 6a). The prognostic value of CRIS-B was independent of tumour stage (Cox regression multivariate analysis, $P < 5 \times 10^{-4}$; Supplementary Table 4) and was maintained in both Duke's B and C patients (Supplementary Fig. 12). This suggests that CRIS-based stratification could be exploited in combination with clinical and pathological parameters for a superior prognostic assessment of CRC. Notably, CRIS-B membership was a negative

predictor of DFS not only in untreated cases (log-rank $\chi^2$, $P < 0.05$, HR = 2.674, CI = 1.045–6.845; Fig. 6b), as previously reported for high stromal content[8,9], but also in cases that underwent adjuvant treatment (log-rank $\chi^2$, $P < 0.005$, HR = 3.042, CI = 1.366–6.777; Fig. 6c). This indicates that the negative prognostic value of CRIS-B is not biased by chemotherapy sensitivity.

To evaluate possible prognostic interactions between the CRIS-B subtype and high levels of CAF infiltration, we stratified tumours with high or low CAF content, as previously estimated in this data set[8] (Supplementary Data 10). By this approach, CRIS-B was found to be informative specifically in low-CAF tumours (log-rank $\chi^2$, $P < 5 \times 10^{-3}$, HR = 2.707, CI = 1.384–5.295; Fig. 6d and Supplementary Fig. 13a). Symmetrically, high CAF content predicted poor prognosis only in non-CRIS-B tumours (log-rank $\chi^2$, $P < 5 \times 10^{-6}$, HR = 5.216, CI = 2.486–10.95; Supplementary Fig. 13b,c). These results suggest that CRIS-B membership and high CAF infiltration identify alternative means to acquire analogous traits of cancer aggressiveness, whose negative prognostic impact is not further exacerbated by the coexistence of the two (Supplementary Fig. 14). This observation prompted us to test a prognostic indicator that integrates CAF score and CRIS-B assignment, whereby all patients categorized as CRIS-B or high CAF, or assigned to both groups, were assumed to have a poor prognosis. The combined classifier performed better than either CAF score or CRIS-B alone (log-rank $\chi^2$, $P < 5 \times 10^{-5}$, HR = 3.248, CI = 1.843–5.724; Supplementary Fig. 15). As previously reported[8], the contribution of the CAF score was negligible for treated patients (Supplementary Fig. 16) and, accordingly, the efficacy of the combined CRIS-CAF predictor was even more pronounced in untreated cases (log-rank $\chi^2$, $P < 5 \times 10^{-5}$, HR = 5.736, CI = 2.334–14.1; Fig. 6e).

Among those individuals who did not receive adjuvant treatment based on clinical and pathological parameters, the integrative deployment of CRIS classes and CAF score was able to discriminate a relatively large subset (30%) of poor-prognosis patients who relapsed in 5 years in more than 40% of cases (sensitivity of 0.68 and specificity of 0.75 for 5-year DFS). The identification of patients with tumours that, in spite of displaying 'benign' clinical and pathological features, appear to have an aggressive biological behaviour, may prove useful to inform a more proactive follow up in such high-risk subpopulation.

Importantly, the association between CRIS-B and poor prognosis was confirmed in an independent cohort of 1,261 samples, which was assembled by combining data from five independent data sets (Supplementary Fig. 17a; Supplementary Data 16). Also in this case, both CRIS-B and high CAF content predicted poor prognosis (Supplementary Fig. 17), but the combination of CRIS-B and high CAF outperformed either of the individual indicators (log-rank $\chi^2$, $P < 1 \times 10^{-7}$, HR = 1.777, CI = 1.433–2.204; Fig. 6f).

As an initial attempt to translate the CRIS taxonomy into a diagnostic tool amenable to clinical applications, we developed a single-sample classifier based on the top scoring pair algorithm (TSP)[41] and its multiclass extension k-TSP[42–44]. A TSP is a binary predictor based on the relative ranking of two measurements (for example, the expression of a pair of transcripts), which switch order between two subclasses of samples. This approach can be extended to multiclass problems by identifying the TSPs associated with each pair-wise subclass comparison and then aggregating the votes across all gene pairs. We took advantage of such method to derive an algorithm for assignment of CRIS subtypes, which we named CRIS-TSP.

To ensure cross-platform portability of the classifier, candidate TSP genes were challenged against a training data set of 624 gene

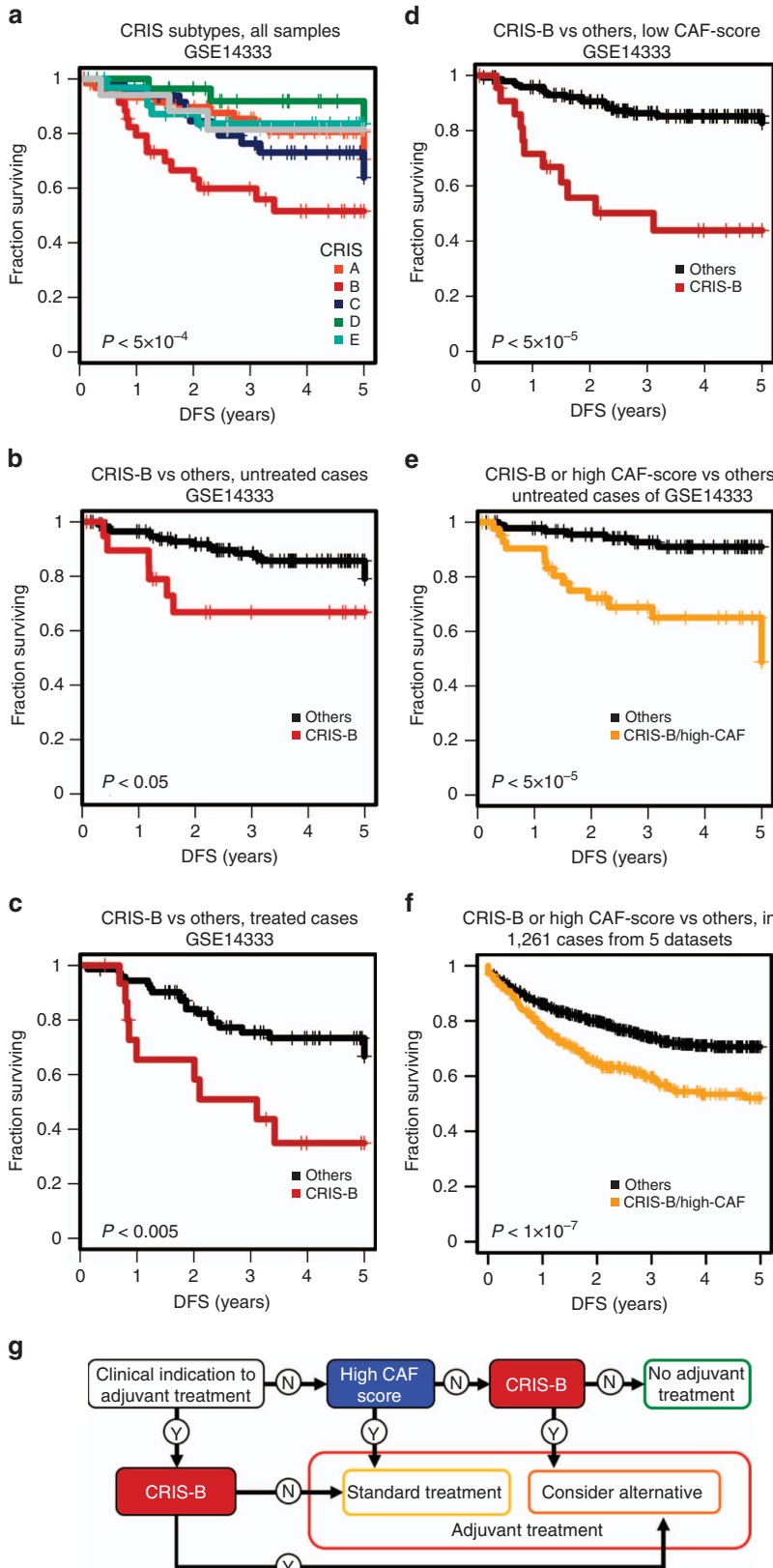

**Figure 6 | CRIS subtypes predict CRC prognosis independent of clinical stage and stromal infiltration.** (**a**) Kaplan–Meier plot of DFS of CRIS subtypes in GSE14333, a prognostically annotated CRC gene expression data set. (**b–e**) Kaplan–Meier plot comparing the DFS of CRIS-B patients versus that of all other patients in cases who did not (**b**) or did receive (**c**) adjuvant chemotherapy and in cases with low (**d**) CAF infiltration. (**e,f**) Kaplan–Meier plot comparing the DFS of cases with high CAF infiltration or CRIS-B membership versus that of all other patients in the untreated population ($n = 138$) of GSE14333 (**e**) and in 1,261 unselected cases extracted from six independent data sets (**f**). (**g**) Flowchart summarizing a possible decision tree taking into account CRIS-B membership and high CAF infiltration, together with the clinical indication to adjuvant treatment, to refine the therapeutic decisional process after surgery.

expression profiles from both PDXs and original tumours, obtained using multiple technological platforms (Supplementary Data 17). This process resulted in the selection of 40 gene pairs (Methods; Supplementary Data 18). When applied to the training set, CRIS-TSP demonstrated good concordance with the original CRIS classifier ($K = 0.7384$). CRIS-TSP was then applied to eight independent data sets, for a total of 2,024 samples (Supplementary Data 17). When challenged against six gene expression data sets annotated for clinical outcome (total samples = 1,487), CRIS-TSP assignments confirmed the poor prognosis of CRIS-B patients (log-rank $\chi^2$, $P < 5 \times 10^{-5}$, HR = 1.6613, CI = 1.307–2.112; Supplementary Fig. 18, Supplementary Data 17). The classification concordance between CRIS and CRIS-TSP was, however, suboptimal in these data sets ($K = 0.6459$). We also found limited concordance when comparing CRIS-TSP classification of surgical resections of liver metastases versus their corresponding PDXs ($K = 0.4053$), overall suggesting that the performance of the CRIS-TSP classifier was less robust when applied outside the training set. To assess whether such reduced classification coherence was due to low classification efficacy of the selected TSP genes, we reconstituted a NTP-based classifier using the same TSP transcripts (CRIS-NTP80, Supplementary Tables 21 and 22). When exploiting the CRIS-NTP80 algorithm for classification of the clinically annotated CRC data sets, the overall classification concordance with respect to the original full-size CRIS classifier was improved over CRIS-TSP ($K = 0.7149$). Consistently, the classification coherence between original tumours and PDXs was also increased ($K = 0.5577$) and similar to that obtained with the full-size NTP. CRIS-NTP80 also confirmed the prognostic significance of CRIS-B (log-rank $\chi^2$, $P < 5 \times 10^{-4}$, HR = 1.581, CI = 1.246–2.006; Supplementary Fig. 18). Altogether, these data show that reducing the size of the CRIS classifier to 80 genes preserves most of its classifying capability across different gene expression platforms, and indicate the feasibility of deploying such a reduced gene set for a single-sample classification based on a TSP approach.

## Discussion

Gene expression analysis based on total RNA of bulk cancer tissues provides an aggregate portrait of the main components that make up the whole tumour ecosystem, including cancer cells, vessels, fibroblasts and immune cells. Although global differences in gene expression patterns have proved useful to distinguish cancer subtypes for effective disease stratification[45], separating the molecular signatures of tissue compartments from measurements of total tumour samples is expected to provide higher resolution of biologically and clinically pertinent parameters[8,46].

The contribution of individual tumour constituents to better capturing some cancer characteristics has been mainly documented for stromal cells in several tumour types. For example, a signature reflecting response of human fibroblasts to serum, suggestive of active wounds, was found in a subgroup of cases at early stages, persisted during treatment, and predicted increased risk of metastasis and death in breast, lung and gastric carcinomas[47]. In pancreatic ductal adenocarcinomas, the integration of tumour- and stroma-specific gene expression profiles resulted in improved prognostic power over traditional signatures[46]. In CRC, stromal traits have been shown to critically impact cancer prognosis and response to therapy[8,9]. On the contrary, how cancer-cell intrinsic gene expression patterns influence subtype classification remains elusive, likely because the proportion of normal tissue lineages present in whole tumour transcriptomes acts as a dominant source of variation that obscures biologically relevant transcriptional features inherently displayed by cancer cells.

To attempt unambiguous exploration of cancer-cell gene expression attributes we took advantage of a large collection of PDXs, in which transcripts of manifest cancer-cell origin could be extracted by the deployment of human-specific probes. The ensuing transcriptional profiles were then leveraged for a class discovery effort. This led to the identification of CRIS, an original classifier that categorizes CRC in five novel transcriptional classes.

Of note, CRIS subclasses only barely overlap with the reported CRC transcriptional classification systems, which empowers a higher dimension of analytical resolution and refines biological insight into CRC heterogeneity. In particular, removal of stromal signals in the class discovery process resulted in remarkable orthogonality between CRIS and the recently published CMS signatures, with lack of classification for CMS subtypes enriched for mesenchymal phenotypes (CMS1 and CMS4) and detection of genetic and functional peculiarities with a potential to instruct novel diagnostic and therapeutic approaches.

CRIS-A is highly enriched for *BRAF*-mutated MSI tumours and *KRAS*-mutated MSS tumours, for which no targeted treatment options are currently available. It is worth noting that such tumours typically exhibit strong glycolytic/hypoxic signatures. Although further studies based on preclinical experimentation and prospective trials in patients are needed to support this assumption, CRIS-A might pinpoint tumour subgroups potentially responsive to anti-metabolic therapies[29]. CRIS-B identifies a previously neglected subset of invasive tumours with poor prognosis and high TGF-β signalling. Since these characteristics are cancer cell specific, CRIS-B is unrelated to the CMS4 mesenchymal subtype, which also includes aggressive tumours and features transcriptional traits of TGF-β pathway activation, but of stromal origin. Novel therapeutic approaches targeting TGFβ-mediated signals in the stroma have been recently proposed[9]; CRIS-B tumours could represent additional candidates for investigational testing of these drugs. CRIS-C categorizes tumours that are strictly dependent on EGFR signals and are sensitive to treatment with anti-EGFR antibodies. The positive predictive value of CRIS-C is of particular importance because it proved to be independent of all known (genetic) biomarkers of response or resistance. CRIS-D groups a subset of tumours enriched for *IGF2* overexpression, which has been recently implicated in desensitization to EGFR blockade in patients with *KRAS* wild-type tumours[35]. The finding that high *IGF2* levels attenuate dependency on the EGFR pathway underscores the functional relevance of this alteration, which is also a candidate target for alternative treatment protocols[35]. Finally, CRIS-E aggregates *KRAS*-mutated, Paneth cell-like CIN tumours that are refractory to treatment with anti-EGFR antibodies. High WNT pathway activity was more generally observed in the CRIS-C–D–E subfamily, thus defining a subset of tumours for which pharmacologic inhibitors of this pathway[48,49] may have therapeutic potential. As a further layer of relevant information for translational purposes, not only does CRIS introduce a new partitioning of known molecular traits, but it also puts forward a number of autocrine signalling loops that are selectively enriched in distinct classes. If validated through functional studies, these signals could constitute an entirely new population of candidate druggable targets for specific CRC subtypes.

Although the analysis of cancer-cell intrinsic traits can provide relevant information for CRC management, the contribution of the stromal compartment should not be overlooked. We and others have reported that the extent of stromal infiltration predicts poor outcome, resistance to radiotherapy and—possibly—sensitivity to chemotherapy[8,9]. Here we show

that the capture of cancer-cell intrinsic traits by CRIS can be efficiently integrated with stromal signatures to obtain even superior prognostic and predictive power. In particular, high CAF score and assignment to CRIS-B independently predict poor prognosis for almost one third of tumours whose clinical and pathological features would not dictate adjuvant treatment. Therapeutically, CAF score and CRIS-B can be integratively employed to stratify CRC patients in three groups: (i) patients with low CAF, non-CRIS-B tumours have good prognosis and may be spared adjuvant treatment; (ii) patients with high CAF-score tumours have poor prognosis but are likely responsive to adjuvant treatment with standard chemotherapy; (iii) patients with CRIS-B, low CAF-score tumours have poor prognosis and are expected to be mostly refractory to conventional adjuvant regimens, but they might benefit from investigational therapies with agents targeting the TGF-β pathway (Fig. 6g). These results call for prospective validation in larger cohorts for drawing definitive conclusions and, if confirmed, could have major clinical implications. The translation of the CRIS taxonomy into a clinically useful companion diagnostic would require the development of a tool for effective classification of individual patients. Here we show that a set of 40 gene pairs amenable to TSP-based single-sample classification retains the classification power of the original 565 gene classifier. However, the performance of CRIS-TSP was negatively affected by retrospective application to existing data sets, likely because of the diversity of procedures adopted and technological platforms used for data generation. Therefore, whenever the goal is to classify already available gene expression data sets obtained by diverse technological platforms (hybridization-based or sequencing-based), NTP-based CRIS categorization remains the option of choice. At the same time, we found that the TSP genes perform well when rechallenged for classification using the NTP approach. This suggests that implementation of this signature into a clinically applicable TSP-based single-sample classifier is feasible for prospective classification of new samples, for which dedicated and standardized data-generation procedures can be adopted.

One potential limitation of our study is that some CRIS features could in fact emerge as a consequence of tumour xenotransplantation. In principle, the PDX approach might exert a number of distortive effects on the transcriptome of cancer cells, including selection drifts related to engraftment and propagation, limited cross-species reactivity between human and mouse cytokines with consequent perturbation of paracrine signals, and lack of proper immune components in recipient animals. However, the likelihood of a strong impact of such biases on the CRIS taxonomy is reduced by the observation that CRIS efficiently classified several data sets from bulk CRC patient tumours, regardless of their source of origin (primary or metastatic). A way to conclusively cope with this issue would be to exploit alternative methods to gather pure cancer cell transcriptional profiles from patient tumours, and test whether the key CRIS features remain valid. Such kind of approaches—mainly based on cell sorting of dissociated tumours or microdissection of histological specimens—have been already applied in small-scale efforts[50–53], but need to be broadened to larger data sets for reliable validation[54].

In conclusion, our data advocate the integrative exploitation of independently assessed cancer-cell intrinsic and stromal CRC transcriptional components as a key opportunity to improve patients' management in the context of precision medicine approaches. Similar findings are beginning to emerge also in other tumour types, using different methodologies[46]. This suggests that the same basic concepts introduced here for CRC can be generalized, with wide impact on cancer diagnosis and treatment.

## Methods

**Specimen collection and annotation.** A total of 244 tumour samples and matched normal samples were obtained from patients who had undergone surgical resection of liver metastases at the Candiolo Cancer Institute, the Mauriziano Umberto I Hospital and the San Giovanni Battista Hospital (Torino, Italy). All patients provided informed consent and study approval was obtained from the review boards of the three institutions (the 'Comitato Etico Istituto di Candiolo—FPO IRCCS', the 'Comitato Etico Azienda Ospedaliera Mauriziano Umberto I' and the 'Comitato Etico Azienda Ospedaliero-Universitaria Città della Salute e della Scienza').

**PDX generation and annotation.** Each collected sample was fragmented and either frozen or prepared for implantation subcutis as previously described[12,55]. Non-obese diabetic/severe combined immunodeficinet mice (4–6 weeks old, both males and females) were used for tumour implantation. At passage two, multiple samples were subjected to gene expression profiling: two samples for 225 tumours, three samples for 13 tumours and four samples for 10 tumours. Whenever possible (149/244 cases, 61%) the different samples analysed originated from independent propagation of regionally distinct areas of the same original tumour, with the aim to take into account intra-tumour heterogeneity. Genetic data and annotation of sensitivity to cetuximab were obtained as described previously[12,36]. In vivo experiments and related biobanking data were stored in the Laboratory Assistant Suite, a web-based, in-house developed data management system for automated data tracking[56]. All animal procedures were approved by the Animal Care Committee of the Candiolo Cancer Institute, in accordance with Italian legislation on animal experimentation.

**Microarray data generation.** RNA was extracted using the miRNeasy Mini Kit (Qiagen), according to the manufacturer's protocol. Synthesis of cDNA and biotinylated cRNA (from 500 ng total RNA) was performed using the IlluminaTotalPrep RNA Amplification Kit (Ambion), according to the manufacturer's protocol. Quality assessment and quantitation of total RNA and cRNAs were performed with Agilent RNA kits on a Bioanalyzer 2100 (Agilent). Hybridization of cRNAs (750 ng) was carried out using Illumina Human 48 k gene chips (Human HT-12 V4 BeadChip). Array washing was performed by Illumina High Temp Wash Buffer for 10 min at 55 °C, followed by staining using streptavidin-Cy3 dyes (Amersham Biosciences). Hybridized arrays were stained and scanned in a Beadstation 500 (Illumina).

**Microarray data preprocessing.** For GSE76402, probe intensity data were extracted using the Illumina Genome Studio software (Genome Studio V2011.1) and subjected to Loess normalization using the Lumi R package[57,58]. To minimize the noise due to cross-species hybridization of transcripts deriving from murine infiltrates in PDX tissues, two pure murine samples were hybridized on human arrays[8] in a pilot experiment, and all probes that generated detectable signals in this assay were removed from further analyses. Moreover, following an initial evaluation of signal quality and distribution, 14 samples were found to be contaminated by murine or human lymphomas and were not further considered in the analysis (9967501046_G, 9967501046_H, 6898368007_A, 6898368007_B, 9235792014_K, 9235792014_L, 8981245030_C, 8803828041_C, 9031292068_C, 9031292069_B, 5688887009_I, 8803828041_D, 8981245029_G, 8981245029_H).

For the remaining 515 samples—derived from 244 unique patients—probes were filtered to select those that showed detectable signal (detection P value = 0) in at least 10% of the samples. For each of such genes, only the probe with the highest variance of signal was selected. Similar analyses were carried out for the panel of liver metastatic CRCs composing GSE73255. The panel included a total of 185 samples corresponding to 167 unique patients.

**Clinical and molecular annotations from the cancer genome atlas (TCGA).** The TCGA data set was obtained from the TCGA data portal, as described previously[8]. Molecular/pathological annotations and IGF2 expression data were downloaded from the Supplementary Tables of the original work[15], while mutational alteration and DFS data were retrieved from the cBioPortal (TGCA, Colorectal Adenocarcinoma Provisional data set; http://www.cbioportal.org/index.do, see Supplementary Tables 1 and 20).

The mutational load was calculated based on exome sequencing (Illumina) data available from the TCGA data portal. In particular the READ and COAD data sets were mined. All the somatic mutations were included in the calculation and normalized assuming an approximate exome size of 30 megabases. The analysis was carried out with the SomaticSignatures R package[59].

To calculate the copy number variation load we took advantage of the segmented data provided by the Broad Institute and available from the TCGA data portal (Genome-Wide Human SNP Array 6.0). Only data unambiguously referred to unique tumour samples were considered. For each sample, all the regions with an absolute segmented value greater than 0.3 were categorized as altered. This threshold was chosen based on previous work, in which standard methods for calling a copy number alteration for a segment in GISTIC analysis of a single sample were defined[60]. The copy number load was calculated as the number of nucleotides included in such altered regions, relative to the sum of all nucleotides in all the

segments identified in the genome of the patient under consideration. The analysis was performed using pandas and scipy packages (http://pandas.pydata.org/; http://docs.scipy.org/doc/scipy/reference/).

To represent gains and losses of whole chromosome arms, we employed the GISTIC[61] results provided by the Broad Institute for the TCGA data set (broad_values_by_arm.txt).

Focal amplifications were obtained from the GISTIC analysis[48] provided by the Broad Institute in the Firehose portal (http://gdac.broadinstitute.org) (Supplementary Data 1).

**Public metastatic and primary colorectal cancer gene expression profiles.** Affymetrix gene expression profiles of 80 metastatic[40] and 290 primary CRCs[24] were downloaded from GEO (GSE5851 and GSE14333, respectively). The data set provided by the Colorectal Cancer Subtyping Consortium, corresponding to GSE39582, GSE2109, GSE17536, GSE13294, GSE20916, GSE37892, GSE33113, GSE13067, GSE35896, GSE23878, GSE5851, PETACC3 and KFSYSCC, was downloaded from the synapse repository. For genes with multiple probe sets, those with the highest average levels were selected.

**Identification of cancer-cell intrinsic transcriptional subclasses.** The identification of cancer-cell intrinsic subtypes was performed by applying unsupervised clustering analysis, following consolidated methods[1]. All the available transcriptional profiles from PDXs were exploited for class discovery. To take into account tumour heterogeneity in the process of subtype identification, PDXs derived from the same original tumour were treated as independent samples. To maximize the portability of the results across multiple platforms, we restricted our analyses only to transcripts that were also explored in the RNAseq data set available from the TCGA data portal. We applied consensus-based NMF[20] to the 1,084 most variable genes. In accordance with Sadanandam et al.[1,62], these were selected by applying a threshold of s.d. = 0.8 followed by sample data normalization $N$ (0,1). NMF was performed with the predetermined number of clusters ($K$) varying from 2 to 6. Quality of the clustering was evaluated by the cophenetic coefficient, with $K = 5$ gathering the highest value and $K = 2$, $K = 3$ displaying similarly high coefficients (Fig. 2a). Among these, $K = 5$ was selected because it (i) demonstrated higher subtyping resolution, splitting $K = 2$ and $K = 3$ in smaller and more homogeneous subclusters (Supplementary Fig. 19a) and (ii) maintained, despite the smaller cluster size, very high cophenetic coefficient also when different s.d. thresholds were used to select the variable genes (Supplementary Fig. 19b).

**Generation of the CRIS classifier.** To identify gene signatures able to discriminate CRIS subtypes, we first focused on the samples that best represented each subtype, using silhouette width—as previously published[1,62]—to remove the samples with negative score values ($n = 90$; Supplementary Fig. 19c) from each cluster. By applying significance analysis of microarrays[21] on the remaining 425 samples, 1,083 genes differentially expressed across subtypes were identified. This was obtained by applying an FDR threshold of 0.005, which was selected to maximize the number of genes to be further selected with PAM[4], while minimizing the overall error rate in cross-validation analyses. Through PAM, we then generated the shrunken centroids of each class by selecting the configuration that minimized the overall error rate in leave-one-out cross-validation analyses. This led to further prioritization of the discriminating transcripts to a total of 903 genes, with an overall error rate of 0.035 (Supplementary Fig. 19d).

For implementation of an NTP-based classifier, we selected genes positively and specifically associated to each of the subtypes. Indeed, the PAM score represents the extent and sign of association of each gene to each class. Starting from the 903 genes selected as specified above, 102 genes did not have a positive PAM score for any of the classes (as a consequence of the centroid shrinkage procedure) and could not be used for NTP. Then, we deployed our published methodology[8] to remove from the classifier those genes for which the major component of signal was defined as having stromal origin. To do so, we calculated the fraction of stromal (mouse) transcripts contributing to the overall signal of each gene using RNAseq data from CRC PDXs, in which mouse stroma substitutes the human stroma[8]. This estimate was exploited to exclude from further analyses 84 genes characterized by a stromal contribution above 50% (ref. 8), which brought to 717 the number of genes to be processed in subsequent steps. The NTP algorithm does not allow redundancy between the signatures used to assign membership to different classes. Thus, all genes featuring a positive PAM score for more than one class had to be non-redundantly assigned to one class only. To do so, we used our previously published procedure[8] and assigned genes that were positively associated to more than one class to the best PAM scoring class only when the second highest value for assignment to another PAM class was at least 0.2 points lower. In all other cases (corresponding to 152 transcripts), the genes were excluded from the analysis. This procedure yielded an NTP-compatible classifier composed of 565 genes, with the following partition: 173 genes for CRIS-A; 73 genes for CRIS-B; 149 genes for CRIS-C; 86 genes for CRIS-D; 84 genes for CRIS-E (Supplementary Data 8). The whole analytical pipeline, from class discovery to the 565-gene NTP classifier, is shown in Supplementary Fig. 20.

**Nearest template prediction.** NTP-based classification[16] was performed using scripts from the GenePattern Bioportal[63] (see URLs). The threshold chosen for significant classification of a sample was Benjamini–Hochberg-corrected false discovery rate (BH.FDR) < 0.2, as previously reported[1]. When referring to published classifications (that is, Fig. 1b, Supplementary Figs 2b,c and 4g), we employed the class assignments provided in the original works by Sadanandam et al.[1] and Guinney et al.[11], respectively.

**Development of a simplified single-sample CRIS classifier.** To develop a simplified classification system for CRIS, we used the TSP approach, a rank-based, parameter-free binary predictor relying on the relative ordering of two features (for example, the order of expression of two genes), and its extension, the k-TSP classifier, which aggregates the votes of multiple TSPs and can be used for multiclass problems[41–43,64] as detailed below.

To this end we first identified candidate genes for classifier development starting from 526 CRIS genes (out of 565) in common across three distinct data sets obtained from different platforms: 254 PDXs analysed on Illumina microarrays, 104 RNA-seq samples from TCGA and 266 samples analysed on Affymetrix microarrays available from the public domain (gse13067, gse13294 and gse35896), for a total of 624 samples (Supplementary Data 17). We then developed our ranked-based classifier using the k-TSP algorithm and compared its agreement with the CRIS classification obtained by the original NTP. None of the 624 samples used to develop our k-TSP-based classifier was included in subsequent analyses investigating the clinical relevance of the CRIS classification. To implement our k-TSP classifier we performed the following analytical steps: (i) identification of the most robust and reproducible set of genes across the three analytical platforms using integrative correlation (ICOR)[65,66]; (ii) splitting of the entire combined data set of 624 samples into training (2/3 of samples) and internal test (1/3 of samples) sets balanced in terms of platform and CRIS class composition; (iii) development of the classification rule using the k-TSP algorithm on the training set; and (iv) assessment of the agreement between k-TSP and NTP using the internal test set.

The TSP algorithm assigns a sample to a specific phenotype if gene A is larger than gene B, or to the other phenotype otherwise. Since there are five CRIS classes, there are ten possible pair-wise comparisons among the CRIS classes: (1) A versus B; (2) A versus C; (3) A versus D; (4) A versus E; (5) B versus C; (6) B versus D; (7) B versus E; (8) C versus D; (9) C versus E; and (10) D versus E.

There are 138,075 possible TSPs that can be formed using all combinations of 526 genes. To avoid over-fitting, however, we limited the search space in the training phase by filtering out all genes that proved to be irreproducible across the three analytical platforms considered (Illumina, RNA-seq, and Affymetrix). To this end, we used the MergeMaid R-package to calculate a gene reproducibility index called ICOR[65,66], which allows to identify genes that are reproducible across distinct data sets without relying on any phenotypic information. We calculated within each separate study, and for each pair of genes, the correlation coefficient of expression value ranks across subjects, and then retained only the genes for which such correlations agreed across studies. Supplementary Figure 21a shows the histograms, the observed and the null distributions (as obtained from 1,000 permutations) for the three pairwise integrative correlations across the three data sets. To select the most reproducible genes we analysed the total integrative correlation obtained by averaging the pairwise integrative correlations using the expectation-maximization (EM) algorithm[67]. This approach allowed us to dichotomize the ICOR values and classify the 526 intrinsic genes based on their reproducibility across platforms. Supplementary Figure 21b shows the distribution of the total ICOR along with the thresholds identified by the expectation-maximization algorithm. In our analyses we retained only the 268 most reproducible genes with mean ICOR > 0.488.

There are still 35,778 possible TSPs that can be formed using all combinations of the 268 most reproducible genes. Hence, to further avoid over-fitting, we limited the search space in the training phase by selecting only the top 50 most differentially expressed genes for each class (if available) using a Wilcoxon rank-sum test, and then by constraining pairing only between genes belonging to the CRIS classes being considered (that is, searching only pairs between CRIS-A and CRIS-B genes when training TSPs to distinguish CRIS-A from CRIS-B, and so on).

We then identified k-TSPs for each of the ten pair-wise comparison between the CRIS classes. To select disjoint TSPs for each class comparison, the genes used to form pairs were omitted from the search in subsequent comparisons. In selecting the most discriminative TSPs we started from the comparisons between CRIS-B and the other classes, since this class showed prognostic value in our previous analyses. We then proceeded to compare CRIS-C to the other classes, since this class showed association with EGFR signalling and cetuximab sensitivity in our previous analyses. We then proceeded with the remaining class comparisons according to the total number of available genes to form the pairs, in increasing order. The order of the search was therefore as follows: CRIS-B versus E, versus D, versus C and versus A; CRIS-C versus E, versus D and versus A; CRIS-E versus D and versus A; CRIS-D versus A. For each of the ten pair-wise comparisons we selected from 1 to 5 TSPs, for a total of 10, 20, 30, 40 and 50 disjoint TSPs, using a total of 20, 40, 60, 80 and 100 non-overlapping genes, respectively.

The classification rule to assign a new sample to a CRIS class was constructed as follows: (i) classify each sample according to each individual TSP included in the

k-TSP classifier and available for analysis (that is, discard TSPs for which the genes were not measured); (ii) count the proportion of votes for each CRIS class (that is, the number of votes for a specific class divided by the total number of comparisons involving that class); (iii) assign each sample to the class with the highest proportion of votes; (iv) in case of ties between three classes or more, do not classify sample ('Unassigned'); (v) in case of ties between two classes, assign samples to the class with the highest proportion of votes in the pair-wise comparisons between these two classes; do not classify sample otherwise ('Unassigned').

In the training phase on 2/3 of the data, we evaluated the concordance between the classification obtained using the k-TSP classifier and the reference NTP classification based on the complete 526 'intrinsic' gene set. We assessed the agreement starting from the minimum set of ten TSPs ($k = 1$), then incremented the number of TSPs until reaching the maximum agreement with NTP. Hence, we developed our kTSP classifier using 10, 20, 30, 40 and 50 non-overlapping TSPs for a total of 20, 40, 60, 80 and 100 genes, respectively. We then compared the agreement of the aggregated k-TSP classifier with NTP classification in the training set and internal testing set and selected the one showing the highest agreement in the internal testing set ($k = 4$ for each pair-wise comparison, for a total of 40 TSPs and 80 genes), as shown in Supplementary Table 5. The 80 genes were also used for NTP-based classification (CRIS-NTP80), by assigning to each CRIS class the 16 TSP genes calling for that class in the pairwise comparisons.

Essential scripts required to apply the CRIS classifiers (NTP and TSP) to independent tumour data sets are provided with the CRISclassifier R package, available as a Supplementary Software.

**SubMap.** The similarity between transcriptional traits underlying subtypes across different data sets was evaluated using the SubMap R package[23] available from gene pattern (http://www.broadinstitute.org/cancer/software/genepattern/).

**Principal component analysis.** Principal component analysis (PCA) was employed to identify the main sources of variation in the TCGA gene expression data set. In particular, we exploited the PCA class from Scikit-learn python library (http://scikit-learn.org/stable/). Enrichment analyses were conducted on the first principal component (PCA0; Supplementary Data 2).

**Comparison of CRC-LM and PDX gene expression profiles.** Genes with high variance (0.8) in the PDX data set were used to calculate Pearson's correlations for any possible CRC-LM/PDX permutation. Correlations of matched versus unmatched CRC-LM/PDX pairs were compared by two-tailed Student's $t$-test.

**Gene set enrichment analyses.** The GSEA software was downloaded from the http://www.broadinstitute.org/gsea/index.jsp. The significance of enrichment was estimated using default settings and 1,000 gene permutations[28,68].

**Sample set enrichment analysis.** For SSEA of curated functional signatures, a score was calculated for each signature and each sample using median-centered Log2 ratios of gene expression values, as follows:

$$\text{Signature score}_j = \overline{\text{ModuleUP}_j} - \overline{\text{ModuleDOWN}_j},$$

where ModuleUP is the average expression value of the genes positively correlated with the phenotype and ModuleDOWN is the average expression value of the transcripts anti-correlated with the phenotype in sample $j$.

We then evaluated the enrichment in class assignment for each CRIS class by performing GSEA preranked analysis using as ranked lists the samples ordered by the score of interest, and as sets the lists of sample membership to the different CRIS subtypes. Calculations were done with 1,000 permutations.

For SSEA of autocrine loops, a total of 472 receptor/ligand interactions were downloaded from the Database of Ligand Receptor Partners (http://dip.doe-mbi.ucla.edu/dip/DLRP.cgi). Then, each set of genes encoding the receptor and its ligands was used to rank samples, based on median-centered Log2 ratios of gene expression values, as follows:

$$\text{Receptor Activation Score}_j = \frac{\text{Receptor}_j + \overline{\text{Ligands}_j}}{2},$$

where Receptor is the mRNA expression of the receptor of interest and Ligands stands for the average expression of its corresponding ligands in sample $j$.

**Statistical analyses.** Statistical analyses were performed by two-tailed Student's $t$-test, Wilcoxon rank test, Fisher's exact test and $\chi^2$-test using the GraphPad Prism software or statistical R packages. For survival analyses, Cox regression hazard model and Kaplan–Meier analyses were conducted using the R-Bioconductor 'survival' package[7,22]. For all tests, the level of statistical significance was set at $P < 0.05$. In case of multiple testing, the results were considered significant when the Benjamini–Hochberg FDR was below 0.2.

**Code availability.** Essential scripts to reproduce our classification on independent CRC data sets can be found in R package CRISclassifier, provided as Supplementary Software to the manuscript. The CRISclassifier package was designed to provide easy and guided access to the full CRIS classification criteria and the TSP-based implementation. Updated versions of this package will be available on Bioconductor (package name: CRISclassifier). Contact claudio.isella@ircc.it for more information.

**Data availability.** Gene expression microarray data generated in the course of this study have been deposited in the GEO database with accession number GSE76402 (PDX data, 529 profiles from 244 patients) and GSE73255 (liver metastases data, 185 profiles from 167 patients).

Other gene-expression data that support the findings of this study are available through the TCGA data portal (TCGAcrcmRNA; URL http://bioconductor.org/packages/release/data/experiment/html/TCGAcrcmRNA.html); from the GEO database (GSE5851, GSE14333; URL https://www.ncbi.nlm.nih.gov/geo/) and from the Synapse data portal from the Colorectal Cancer Molecular Subtyping Consortium (GSE39582, GSE2109, GSE17536, GSE13294, GSE20916, GSE37892, GSE33113, GSE13067, GSE35896, GSE23878, GSE5851, PETACC3 and KFSYSCC; URL http://sagebase.org/research-projects/colorectal-cancer-subtyping-consortium-crcsc/).

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

## Acknowledgements

We thank our friends and colleagues of the Laboratories of Translational Cancer Medicine and Oncogenomics for suggestions and discussion; Alessandro Ferrero, Francesco Giraldi, Alfredo Mellano, Andrea Muratore, Mauro Papotti, Gianluca Paraluppi and Anna Sapino for sample acquisition; and Antonella Cignetto, Daniela Gramaglia and Francesca Natale for secretarial assistance. Work in the authors' laboratory was supported by AIRC, Associazione Italiana per la Ricerca sul Cancro, Investigator Grants 15571 (A.B.), 12944 (E.M.), and 14205 (L.T.). AIRC 2010 Special Program Molecular Clinical Oncology 5 × 1,000, project 9970 (E.M. and L.T.). 2012 Fight Colorectal Cancer—AACR, American Association for Cancer Research, Career Development Award in memory of Lisa Dubow, Grant Number 12-20-16-BERT (A.B.). MIUR FIRB, Ministero dell'Università e della Ricerca, Fondo per gli Investimenti della Ricerca di Base—Futuro in Ricerca (A.B.). Fondazione Piemontese per la Ricerca sul Cancro-ONLUS, 5 × 1,000 Ministero della Salute 2010 (E.M.) and 2011 (E.M. and L.T.). Transcan, TACTIC (L.T.). NIH grant P30CA006973 (L.M.). A.B., E.M. and L.T. are members of the EurOPDX Consortium.

## Author contributions

C.I. contributed study design, data analysis and manuscript writing. R.P., F.G., C.P. and E.Z. collected samples and generated microarray data. F.G., E.Z. and F.O. generated and curated PDX molecular annotations. F.B., S.E.B., L.M. and A.F. contributed bioinformatics analyses. R.S. interpreted histological analyses. C.B., E.F. and L.T. contributed study design and manuscript writing. A.B. and E.M. contributed study design, data analysis, bioinformatics, manuscript writing and project oversight.

## Additional information

**Competing interests:** The authors declare no competing financial interests.

**Publisher's note**: 

