## [Peer review file · Nature Communications]

Reviewers' comments:

Reviewer #1 (Remarks to the Author):

In the manuscript by Isella et al., the authors have defined a new classification system of colorectal cancer, based in intrinsic gene expression features of cancer cells. This has been accomplished by analyses of PDX models, where only the cancer cells are of human origin. Expression profiles from paired liver metastatic samples were included, samples which contain an admixture of liver, stromal, and blood cells. The idea here is that the intrinsic expression of the cancer cells can be deduced from the PDXs. The rationale for development of a cancer-cell intrinsic signature was observations made by the authors of classification incongruence between PDXs and their original counterparts. Depletion of stroma-derived signals is claimed to be the major source of transcriptional variation between surgical specimens and PDXs (fig 4g).

The new CRIS-classification may also be important towards use in personalized medicine for colorectal cancer in that the classes CRIS C & A predict sensitivity and resistance, respectively, to the EGFR inhibitor cetuximab. The authors have done a good job of describing and proving this, including that the predictive power is independent of KRAS/NRAS mutated cases.

Overlap with "consensus molecular subtype" (CMS) classification is poor, in particular in the CMS-class which corresponds to samples with high levels of cancer associated fibroblasts. Overlap with MSI is good (two CRIS-classes include virtually all MSI cases; fig 3c). Data is also shown of CRIS classes giving less overlap with phenotypes such as CIN and CIMP.

The authors state that only human RNA will bind to the microarrays, there is, however, substantial evidence of cross hybridisation in microarray experiments. The authors solved this by subtracting features with expression signals from hybridisations of pure murine samples onto the human arrays from further analyses.

All-in-all, I find this study and manuscript of high interest and that it has high standards of the analyses and presentation, including figures and language.

The manuscript may further be improved by some of the following suggestions:

1. Cell lines: I further find it clever to compare PDX models with their own paired liver metastatic tumour samples, and the authors have done a substantial job to get a high number of PDX models generated and analysed (n=515). Although the number of PDXs is high, effort must be put into the validation of the subtyping in sufficient number of datasets and samples. The authors have tested several datasets for various validation purposes. However, more emphasis into validation in yet more available datasets should be made. There are several available datasets that the authors could make use of for validation of the subtyping. Both primary and metastatic cancers. The intrinsic expression should also be readily seen from cell lines (data available from e.g. Cancer Cell Line Encyclopedia project), and then the authors should both check this, and e.g. implement into Figs 1 & 2. Also, list commonly used CRC cell lines belonging to each of the CRIS-classes.

2. More emphasis/discussion on how applicable the CRIS subtypes are to RNA-seq data.
3. Also, I would like to have more description of the practicalities on how the research and clinical community can implement the subtyping to own data. This should both included data from various microarray, RNA-seq, RT-PCR based (if applicable) platforms. And also how this would apply to data from clinical cohorts, PDXs and cell lines.
4. I would have liked a bit more critical discussssion that the PDX context is not only giving the true or intrinsic cancer signal, but here, the cancer cells (and thus the expression patterns) are also influenced by the murine context. Would it also have been an option to rather dissociated cells from bulk, and sort out the cancer cells? Are there already such a dataset available? In any case, the opportunity this would give should be discussed.
5. Spelling misstake in Figure legend 2: "K= 2-to-5" should be "K= 2-to-6". In the same fig legend; if the TCGA data here is from RNA-seq (and not microarrays), then this fact should be mentioned here (in the legend).
6. Figure 6e. In the high CAF infiltration, the CRIS classification seem to perform poorly. This should be better discussed.

Reviewer #2 (Remarks to the Author):

In the manuscript entitled "A global gene expression analysis of cancer-cell intrinsic transcriptional traits defines a new classification system for colorectal cancer with improved biological resolution and superior predictive and prognostic value" Isella and colleagues describe a new taxonomy for colorectal cancer (CRC) based on 5 newly discovered CRC intrinsic subtypes (CRIS). This taxonomy is obtained in patient derived xenografts (PDXs), hence it is based on cancer cells gene expression profiles (GEPs) that are not masked or "contaminated" by stromal contributions. The authors base their study on their previous observation that currently known CRC molecular subtypes are largely driven by GEPs from cancer associated fibroblasts (CAFs). The authors further characterize the newly discovered CRIS using several approaches, identifying biological themes associated with each class, and correlating these subtypes with distinct prognosis and response to cetuximab in public domain data.

The study is novel and leverages an extraordinary resource: a collection of over 500 transcription profiles derived from ~250 CRC patients. The release of such resource in the public domain as the result of the publication of this study will definitely benefit and positively impact the CRC research field. Furthermore, the newly discovered taxonomy might prove useful in understanding CRC biology, and in implementing strategies for improved clinical decision making. The enthusiasm for this study is diminished by the fact that the authors do not address the translational aspects of their discovery and do not fully develop their taxonomy into a tool for clinical use. It is not clear, indeed, how the CRIS classification will be used to assess patient prognosis or decide about treatment. The taxonomy is indeed based on more than 500 genes and the authors do not make any attempt to simplifying it to a manageable size and to implementing it as a clinical tool (for instance as it has been done with the breast cancer intrinsic subtypes and the PAM50 classifier). Another weakness is the fact that the prognostic and predictive value of this taxonomy is only investigated and assessed in two datasets despite the availability of many more cohorts from the public domain. Failing to do this substantially reduced the enthusiasm for and the significance of this study, making it unsuitable for publication on Nature Communications in its current status.

The methodology and the design used to perform this study are somewhat complex to follow. The authors used a multi-step procedure to 1) discover, 2) refine, and 3) validate a gene expression signature system associated with distinct molecular CRC subtypes. To achieve this they combined unsupervised class discovery techniques with supervised class comparison and prediction methods. Throughout the study several parameters have been tuned, and these choices have not been completely justified. Similarly in some instances simpler and more straight forward approaches could have been used. The fact that this approach is published (it is basically the same one used by Sadanandam et al) does not make it less cumbersome. Nevertheless, findings are in general very well presented and figures are clear and consistent in using colors codes. A number of issues that should be addressed are discussed below.

In the study, although the analytical design is overly complicated, statistics is usually used appropriately. Generally each time a p-value is reported in the text the authors also reported

the corresponding statistical test that was used to obtain it. The authors, however, should also assess whether there are deviations from normality and hence whether non parametric tests are more appropriate. Furthermore, in light of the recent statement from the American Statistical Association (<http://www.amstat.org/newsroom/pressreleases/P-ValueStatement.pdf>) it would be advisable to report effect sizes too. More comments on use of statistical tests is included in the detailed comments below.

Detailed comments:

1) Results Section: "CRC PDXs fail assignment to public transcriptional subtypes".

The authors fail to assign to public transcriptional subtypes CRC samples from their PDXs cohort. They speculate that this is due to the lack of stromal contribution. Although likely and plausible, more efforts should be devoted to prove this point. To support this hypothesis, indeed, the authors simply report the classification incongruence and the fact that PDXs are more similar to their matched tumors. Such findings prove that different results are obtained in PDXs and in tumors, and that matched pairs of PDXs and tumors are more similar to one another than random pairs. These results do not prove that this discrepancy is due to the absence of stromal contributions to GEPs. Furthermore:

a) For the loss of classification rate in SSM classes the authors report p-values from chi square tests, however, it is not clear how chi-squared tests were used for this purpose. What is the null hypothesis tested?

b) For the difference in correlation between matched or random pairs the authors used t-tests and hence they should make sure that such correlations are normally distributed.

2) Results Section: "Cancer-cell intrinsic transcriptional traits classify colorectal cancer".

The procedure used to develop the CRIS taxonomy is not extensively explained and a flow-chart summarizing the procedures and steps involved would be beneficial. This is particularly important since the combination of unsupervised and supervised learning methods used is complex and somewhat cumbersome. In addition to this:

a) The authors used 515 PDXs or so to learn the new taxonomy. However there are only 225 patients and it is not clear how such redundancy was handled.

b) The authors have a matched set of tumor-PDXs pairs. It appears that they developed the taxonomy using all the PDXs and then compared results between PDXs and tumors in the matched subset. If so, the classification performance in tumors is somewhat "inflated", because matched cases are correlated. The authors could perhaps make a better use of their data by learning the taxonomy on the cases that are not matched, applying it then to the matched PDXs and tumors.

c) The authors used non-negative matrix factorization (NMF) with 5 different initial conditions for the factorization algorithm and with k varying from 2 to 6. They identified an optimal partitioning into 5 clusters. Is this a real optimum, or other cluster numbers are similarly plausible (cophenetic coefficients are indeed really similar for k=2 and k=5)? How a different number of cluster would affect downstream analyses (e.g., association with functional processes, association with prognosis and response to therapy, and the likes)? In the end on TCGA data the 5 classes are clustered into two major groups... Or clustering, perhaps, is not

an ideal tool...

d) Furthermore, NMF was performed using the most variable genes ($SD=0.8$). What is the rationale for selecting an SD equal or larger than 0.8? How many genes were retained? Why not to use $SD=1.5$? Some rationale should be provided supporting this choice. As alternative the authors should show that this gene filtering procedure does not affect the number of clusters identified in downstream analyses.

e) The authors focused on samples best representing each subtype based on silhouette width removing samples with negative score values. How much the results would change using more strict (or relaxed) criteria for inclusion of the samples in subsequent analytical steps?

f) The author used Significance Analysis of Microarrays (SAM) on 425 "good" samples, identifying 1083 genes differentially expressed across subtypes with FDR of 0.005. Similarly to what said before, how much the results would change using more strict (or relaxed) criteria for gene selection using SAM in subsequent analyses? Why did the authors used an FDR of 0.005 and not 0.1 or 0.001? How much choosing an alternative threshold would change downstream results?

g) Prediction Analysis for Microarrays (PAM) was applied to develop shrunken centroids for each class, further reducing the number of genes used to 903 genes, with an overall error rate in leave one out cross-validation of 0.035. Is this final set of 903 genes resulting from minimizing the cross-validation error in PAM? If so the authors should simply state it.

Similarly to what observed above about NMF and SAM, also in this case the authors should prove that the parameters that they used did not affect downstream findings.

h) To generate a cancer-cell intrinsic classifier not influenced by stroma-derived transcripts, the authors further selected 565 genes with unambiguous epithelial expression. This allowed to confidently classify 94% of the samples in the PDX training dataset. How these epithelial genes were identified? Why exactly 565? How did they "confidently" identified the 94% of the samples? Which gold-standard did the authors used? Isn't the PDXs set the training set? Shouldn't all samples from the training set be correctly classified?

i) In general, the gene filtering/selection procedure is not very well described and clarifications are need. In the methods section the authors state that "If a gene was positively associated to more than one class, it was assigned to the best scoring class only when the second highest value was at least 0.2 points lower. In all other cases, the gene was excluded from the analysis." How many genes were filtered because of an ambiguous association with multiple clusters? How was the 0.2 threshold selected? How much this threshold affects the classification? Is there a reason to use only disjoint sets to classify patients?

j) The authors also state that "the genes with an estimated stromal contribution above 50% were removed from the analysis." How was this "stromal contribution" assessed? How is stromal contribution quantified? What is the rationale for using 50% as the filtering threshold? How this impact on downstream analyses?

k) At page 5 the authors state "All classes were well represented in both the PDX and CRC-LM datasets, displayed similar distributions, and were sustained by similar underlying transcriptional traits". It is not clear what this really means, the authors should elaborate more on this.

l) The authors tested their CRIS classifier on two independent gene expression datasets of primary CRCs (TCGA and GSE14333). Why did they use only these 2 datasets (there are many more available)?

3) Results Section: "Major peculiarities of CRIS classes".

- a) The authors showed that MSI and MSS samples are not equally partitioned across CRIS subfamilies in the TCGA dataset. Is this difference significant as it appears from the figure (especially between CRIS-A and the rest of the CRIS groups)?
- b) The authors stated that "These two classes proved to be also enriched for right-colon tumors featuring mucinous histology, a hyper-mutator genotype, and a CpG island methylator phenotype". From what this reviewer can understand from Figure 3D the enrichment of samples with these characteristics is in CRIS-A compared to CRIS-B, CRIS-D, and CRIS-E, not compared to CRIS-C. A chi-squared test could be used to assess whether these differences are significant or not.
- c) The authors described several copy number alterations associated with CRIS in TCGA data - e.g., gains of entire chromosome arms in CRIS-C (Chr8p, Chr20p and Chr20q) and CRIS-E (Chr13q), focal amplification of 8q.24.21 in CRIS-C, etc... From Figure 4a, Chr13q clearly appears to be enriched also in CRIS-C, and perhaps also in CRIS-D. Furthermore additional red "bands" are evident in Figure 4a in other classes, hence it is not clear what is the rationale to focus on the selected regions. How was this enrichment defined? Did the authors set a cut-off to define enrichment, then did they count the enrichment events in a given genomic region, and then finally compare proportions across CRIS classes? Or is this analysis simply exploratory?
- d) In the methods section the authors state: "For each sample, all the regions with an absolute segmented value greater than 0.3 were categorized as altered." Is there any rationale for selecting the 0.3. threshold?
- e) Furthermore they state: "The copy number load was calculated as the number of nucleotides included in such altered regions, relative to the sum of all the nucleotides covered by the sample segments." The authors should elaborate more about such "copy number load". Indeed, while it is clear what the authors mean by "number of nucleotides in the altered regions", it is not clear what they mean by "sum of the nucleotides covered by the sample segments". All the segments identified in the genome of the patient under considerations? The segments in the region for that patient? Clarifications are needed here.
- f) Always in the method section the authors state: "All the regions scoring outside of the noise thresholds defined by GISTIC (0.1 and -0.1) were categorized as altered." How selecting this or another threshold might affect the results?
- g) The authors compared mutation prevalence in CRIS classes and report Fisher's exact test p-values. How did the authors perform the Fisher's exact tests? Did they compare one CRIS class to the remaining ones lumped together? This applies for BRAF, KRAS, and TP53.
- h) Furthermore, how did the authors test for enrichment and depletion (as described for TP53)? Did they perform each pair-wise comparison between the CRIS classes? If so, did they correct for multiple testing?
- i) To functional characterize the CRIS classes the authors use a combination of approaches based on GSEA. At page 7 they state: "we ranked samples by scores calculated from manually curated signatures relevant for CRC biology, and employed the GSEA algorithm to test each CRIS subtype for enrichment in high-rank samples." This statement is really confusing. Which manually curated genes sets were used and how were they selected? How were the high rank samples selected? What does this enrichment ultimately represent?
- j) The authors further "estimated the enrichment of each subtype in a number of mitogenic/anti-apoptotic autocrine loops by ranking the samples based on concomitant

ligand-receptor expression." This is an interesting approach. The authors might want to consider the manuscript by Boca et al, in which samples-centered enrichment was introduced for the first time (PMID:21092299) as an example of the procedures one should put in place and use to assess the reliability and robustness of a new test/approach.

k) Enrichment analyses can only show what it is represented by the functional classes (i.e., the gene sets) selected and fed in the enrichment test itself. How much bias is introduced by this approach in selecting "relevant CRC gene sets"? Why the authors did not use an unbiased approach based on a larger and comprehensive collection of gene sets (for instance as available from MSigDB)? Finally, how did the authors handle multiple testing resulting from the use of multiple gene sets (if they actually performed a test to identify the most strongly associated processes)?

l) Based on the radar plots the authors state that "Both enrichment analysis and phenotypic signatures were concordant in attributing secretory and MSI-like features to CRIS-A, in agreement with the mucinous histology of this subgroup detected in the TCGA dataset (Fig.4E)." Figure 4E is, indeed, very effective in showing enrichment across different functional gene sets for each CRIS class identified, however the authors should investigate the relationships among the different gene lists used to make sure that such enrichment is driven by different genes and not by a common core set of genes shared by the different lists used for the analysis.

m) The authors reported that "blinded pathological inspection of images from TCGA samples revealed that CRIS-B members included a large number of poorly differentiated tumors, in which the glandular architecture of the original tissue was completely lost or barely detectable." Is this association anecdotal or there is a significant enrichment of undifferentiated tumors in the CRIS-B group?

n) In general, are the different functional and phenotypic characteristics associated with the CRIS classes quantifiable in some way? Did the author quantify somehow all these associations? Are the gene sets represented in Figure 4E and 4F all those that were investigated or only those enriched among all the investigated ones? In the figure legend a p-value from a Fisher's exact test is mentioned, however it is not really clear where and how this test was used.

o) The authors show that there is limited overlap between sample partition based on CRIS and CMS. Can the authors test for the lack of the association? Furthermore, Figure 4G is mapping samples between CMS subtypes and CRIS groups, however, cross-tabulation would probably show this relationship more effectively and in a quantitative way.

p) At page 8 the authors state: "As shown by the heatmaps in Fig. 4G, asymmetric distribution of stromal scores was clearly detected in the CMS partition but not in the CRIS partition". The figure clearly shows that CAFs genes are spread over different CRIS groups, while these are basically all grouped in CMS4. It is not clear, however, what the author means by "stromal score" here... Perhaps this lack of clarity is due to space constraints, but an effort should be made to clarify what these heatmaps exactly show.

4) Results Section: "CRIS predicts response to anti-EGFR antibodies"

The authors used the GSE14333 dataset to study the association of CRIS classes to anti-EGFR antibodies response. They have shown that CRIS-C subtype is over-represented in the cetuximab-sensitive patients, and that the opposite happened for CRIS-A. These associations were investigated using cross-tabulation and Fisher's exact tests, reporting odds ratio and p-values. This association persisted also after removing from the analysis the

cases harboring genetic alterations conferring resistance. The authors also investigated a set of genes indicative of EGFR pathway activity, which showed higher expression in CRIS-C tumors.

- a) The authors should specify to which group CRIS-C patients were compared by t-test when they evaluated EGFR pathway activity. The other classes lumped together? Furthermore, is the use of the t-test appropriate (normally distributed data)?
- b) The authors used multiple linear regression and demonstrated that genetic markers, CRIS classes, and EGFR pathway activity have independent predictive value. It is not clear, however, how the gene expression profile for EGFR pathway activity was summarized for each patient in order to fit the regression. Furthermore, the authors did not clarify which is the response variable used in such regression model. If the response to treatment is the outcome, how was this measured? Is the response sensitivity to treatment? If so, isn't the outcome binary and a logistic model should be used instead of a linear regression?
- c) To explore CRIS prognostic impact the authors "examined the association of CRIS subtypes with disease-free survival (DFS) of 290 patients included in a publicly available and clinically annotated CRC dataset (GSE14333)". Aren't there more datasets with survival information available? More effort should be put into this analysis to strengthen these prognostic associations. Aren't survival data also available for TCGA?
- d) At page 10 the authors state: "When challenged in tumors with high or low stromal content, as estimated by a transcriptional CAF score, CRIS-B was informative specifically in low-CAF tumors (Fig. 6D and E; log rank chi square, $P < 5 \times 10^{-5}$). These results prompted us to test a prognostic indicator that integrates CAF-score and CRIS-B assignment in an "or-based" algorithm. The combined classifier performed better than either CAF score or CRIS-B alone". How does this combined classifier work? No details about the classification rule employed were found in the manuscript.
- e) The authors further state: "the integrative deployment of CRIS classes and CAF score was able to discriminate a relatively large subset (30%) of poor-prognosis patients who relapsed in 5 years in more than 40% of cases (sensitivity of 0.68 and specificity of 0.75 for 5-year DFS)". How about comparing classifications results with CRIS, CAF, and the combined classifiers using AUC? How about using more than just one dataset?

5) On reproducible research. The importance of reproducible research and data sharing has been strongly remarked by multiple Institutions and Publishers (see for instance the US Institute of Medicine report at <http://www.nature.com/news/lapses-in-oversight-compromise-omics-results-1.10298>). For this reason, and in accordance to recommendations about reproducible research (see "Reproducible research in computational science." by Peng, R.D. in *Science* 334, 1226-1227, 2012), all the scripts and code used for performing the analyses should be provided as supplementary material. To this end both the PDXs and tumor datasets used in this study (GSE76402 and GSE73255) are not currently available and hence none of the analyses performed could be reproduced and evaluated by this reviewer. For a true assessment of the validity of the findings presented in this study a link to these datasets should have been provided to the reviewers.

Point-by-point reply to reviewers' comments:

Reviewer #1:

"In the manuscript by Isella et al., the authors have defined a new classification system of colorectal cancer, based in intrinsic gene expression features of cancer cells. This has been accomplished by analyses of PDX models, where only the cancer cells are of human origin. Expression profiles from paired liver metastatic samples were included, samples which contain an admixture of liver, stromal, and blood cells. The idea here is that the intrinsic expression of the cancer cells can be deduced from the PDXs. The rationale for development of a cancer-cell intrinsic signature was observations made by the authors of classification incongruence between PDXs and their original counterparts. Depletion of stroma-derived signals is claimed to be the major source of transcriptional variation between surgical specimens and PDXs (fig 4g).

The new CRIS-classification may also be important towards use in personalized medicine for colorectal cancer in that the classes CRIS C & A predict sensitivity and resistance, respectively, to the EGFR inhibitor cetuximab. The authors have done a good job of describing and proving this, including that the predictive power is independent of KRAS/NRAS mutated cases.

Overlap with "consensus molecular subtype" (CMS) classification is poor, in particular in the CMS-class which corresponds to samples with high levels of cancer associated fibroblasts. Overlap with MSI is good (two CRIS-classes include virtually all MSI cases; fig 3c). Data is also shown of CRIS classes giving less overlap with phenotypes such as CIN and CIMP.

The authors state that only human RNA will bind to the microarrays, there is, however, substantial evidence of cross hybridisation in microarray experiments. The authors solved this by subtracting features with expression signals from hybridisations of pure murine samples onto the human arrays from further analyses.

All-in-all, I find this study and manuscript of high interest and that it has high standards of the analyses and presentation, including figures and language."

→ We thank the reviewer for his/her positive comments and fruitful suggestions, which have been carefully addressed, as detailed in the point-by-point replies below. For the reviewer's convenience, all changes have been marked in red in a dedicated version of the manuscript (Isella et al_Manuscript_Marked Text).

"The manuscript may further be improved by some of the following suggestions:"

1. *"Cell lines: I further find it clever to compare PDX models with their own paired liver metastatic tumour samples, and the authors have done a substantial job to get a high number of PDX models generated and analysed (n=515). Although the number of PDXs is high, effort must be put into the validation of the subtyping in sufficient number of datasets and samples. The authors have tested several datasets for various validation purposes. However, more emphasis into validation in yet more available datasets should be made. There are several available datasets that the authors could make use of for validation of the subtyping. Both primary and metastatic cancers. The intrinsic expression should also be readily seen from cell lines (data available from e.g. Cancer Cell Line Encyclopedia project), and then the authors should both check this, and e.g. implement into Figs 1 & 2. Also, list commonly used CRC cell lines belonging to each of the CRIS-classes."*

→ In the original manuscript we evaluated the performance of CRIS on approximately 700 gene expression profiles extracted from 3 independent CRC datasets from both primary tumors and liver metastases. Now, we have extended the analysis to a much larger collection of samples. Collectively, CRIS has been newly validated on a total of 3738 CRC samples retrieved from 15 independent datasets (also including a dataset of 151 CRC established cell lines); of these, 3396 (90.8%) have been confidently classified by CRIS (FDR < 0.2) through NTP. The gene expression profiles of the additional datasets examined were obtained using different technologies, including Affymetrix, Illumina RNAseq, and Illumina BeadArrays. This further indicates that the CRIS classifier can be successfully applied across different technologies and experimental settings. These data are now presented in the Results (page 6), Figure 2i, and Supplementary Table 11.

2. *“More emphasis/discussion on how applicable the CRIS subtypes are to RNA-seq data.”*

→ We thank the reviewer for the suggestion. The Results and Discussion sections of the manuscript (pages 5-6 and page 15, respectively) have now been modified to highlight that the classifier was challenged against both hybridization-based and sequencing-based transcriptional profiles, with high classification performance.

3. *“Also, I would like to have more description of the practicalities on how the research and clinical community can implement the subtyping to own data. This should both included data from various microarray, RNA-seq, RT-PCR based (if applicable) platforms. And also how this would apply to data from clinical cohorts, PDXs and cell lines.”*

→ To comply with the reviewer’s suggestion and facilitate the fruition of CRIS subtyping by the community of basic researchers and clinicians, we have adopted a two-pronged strategy:

- 1- As an initial attempt to translate the CRIS taxonomy into a diagnostic tool amenable to clinical applications, we developed a single-sample classifier based on the top scoring pair algorithm (TSP) and its multiclass extension k-TSP. A TSP is a binary predictor based on the relative ranking of two measurements (e.g., the expression of a pair of transcripts), which switch order between two subclasses of samples. This approach can be extended to multiclass problems by identifying the TSPs associated with each pair-wise subclass comparison and then aggregating the votes across all gene pairs. We took advantage of such method to derive an algorithm for assignment of CRIS subtypes, which we named CRIS-TSP. To ensure cross-platform portability of the classifier, candidate TSP genes were challenged against a training dataset of 624 gene expression profiles from both PDXs and original tumors, obtained using multiple technological platforms (Supplementary Table 21). This process resulted in the selection of 40 gene pairs (Methods, pages 19-21; Supplementary Table 22). When applied to the training set, CRIS-TSP demonstrated good concordance with the original CRIS classifier ($K = 0.7384$). CRIS-TSP was then applied to eight independent datasets, for a total of 2024 samples (Supplementary Table 21). When challenged against 6 gene expression datasets annotated for clinical outcome (total samples = 1487), CRIS-TSP assignments confirmed the poor prognosis of CRIS-B patients (log rank chi square, $P < 5 \times 10^{-5}$, HR= 1.6613, C.I. = 1.307 - 2.112; Supplementary Fig. 18, Supplementary Table 21). The classification concordance between CRIS and CRIS-TSP was, however, suboptimal in these datasets ($K = 0.6459$). We also found limited concordance when comparing CRIS-TSP classification of surgical resections of liver metastases versus their corresponding PDXs ($K = 0.4053$), overall suggesting that the performance of the CRIS-TSP classifier was less robust when applied outside the training set. To assess whether such reduced classification coherence was due to low classification efficacy of the selected TSP genes, we reconstituted a NTP-based classifier using the same TSP transcripts (CRIS-NTP80, Supplementary Tables 21 and 22). When exploiting the CRIS-NTP80 algorithm for classification of the clinically annotated CRC datasets, the overall classification concordance with respect to the original full-size CRIS classifier was improved over CRIS-TSP ($K = 0.7149$). Consistently, the classification coherence between original tumors and PDXs was also increased ($K = 0.5577$) and similar to that obtained with the full-size NTP. CRIS-NTP80 also confirmed the prognostic significance of CRIS-B (log rank chi square, $P < 5 \times 10^{-4}$ HR= 1.581, C.I. = 1.246 - 2.006; Supplementary Fig. 18). Altogether, these data show that reducing the size of the CRIS classifier to 80 genes preserves most of its classifying capability across different gene expression platforms, and indicate the feasibility of deploying such a reduced gene set for single sample classification based on a TSP approach. These data also point to NTP-based CRIS categorization as the option of choice, whenever the goal is to classify already available gene expression datasets obtained by diverse technological platforms. Implementation of a clinically applicable TSP-based single-sample classifier is feasible for prospective classification of new samples, for which dedicated and standardized data-generation procedures can be adopted. These new observations have been included in the Results, pages 12-13, and Discussion, page 15.
- 2- To facilitate exploitation of CRIS classification by the community of researchers and clinicians, we have implemented an R package called CRISclassifier (included as a supplementary file: Isella et al_CRISclassifier_R-package), which exploits NTP for a nearest-neighbor and TSP classification approach to CRIS-categorize samples in datasets. Through this tool, virtually any gene expression dataset can be classified by applying the exact same procedures that have been adopted in this study. To assist non-expert users, a document reporting step-by-step instructions to run the pipeline has been included in the package.

4. *“I would have liked a bit more critical discussion that the PDX context is not only giving the true or intrinsic cancer signal, but here, the cancer cells (and thus the expression patterns) are also influenced by*

the murine context. Would it also have been an option to rather dissociated cells from bulk, and sort out the cancer cells? Are there already such a dataset available? In any case, the opportunity this would give should be discussed."

→ We thank the reviewer for the suggestion and agree that the biological implications of our approach should be better put in context. We have therefore added a new section in the Discussion in which we elaborate on the limitations of analyzing tumor-stroma cross-talks in the PDX setting. Further, we now put forward potential alternative approaches to investigate the contribution of stromal versus cancer-cell transcripts to tumor behavior (page 15).

We concur that the interaction with mouse stroma could impact the transcriptional wiring of cancer cells in ways that are not necessarily analogous to those of the human stroma. There are multiple reasons for this, the most important being the possibly limited cross-species reactivity of some paracrine signaling activities and the lack of host immunoreactivity. However, if CRIS classes were strongly biased by transcriptional drifts (or artifacts) imputable to mouse xenografting, one would expect that the CRIS classifier does not confidently classify surgical specimens of tumor bulks directly taken from patients. This turned out not to be true. Indeed, we were able to classify more than 3500 CRC transcriptional profiles, extracted from 16 independent datasets obtained with 3 different technological platforms, with an average confidence of more than 90% overall (Figure 2i).

We also concur with the reviewer that the possibility to isolate cancer cells from stromal components is a reasonable alternative approach to test the same working hypothesis. We now cite in the Discussion some proof-of-concept attempts that have been pursued in this direction (Calon et al. 2012, Kagawa et al. 2013, Dunne et al. 2016). However, to our knowledge, the multiplicity of the datasets generated by such means remains limited. This could reflect the logistical difficulties that are faced when physical sample dissociation is required as a preliminary preparation step before transcriptional profiling. Although the scale of published works is too small to enable class discovery approaches, the potential merit of cancer vs. stroma physical separation as an alternative validation of PDX-based CRIS subtyping is now mentioned in the Discussion (page 15).

REFERENCES:

- Calon A. et al. Dependency of colorectal cancer on a TGF- β -driven program in stromal cells for metastasis initiation. *Cancer Cell* **22**, 571-84 (2012).
- Kagawa Y. et al. Cell cycle-dependent Rho GTPase activity dynamically regulates cancer cell motility and invasion in vivo. *PLoS One* **8**, e83629 (2013).
- Dunne P. et al. Challenging the Cancer Molecular Stratification Dogma: Intratumoral Heterogeneity Undermines Consensus Molecular Subtypes and Potential Diagnostic Value in Colorectal Cancer. *Clin Cancer Res* **22**, 4095-104 (2016).

5. "Spelling mistake in Figure legend 2: "K= 2-to-5" should be "K= 2-to-6". In the same fig legend; if the TCGA data here is from RNA-seq (and not microarrays), then this fact should be mentioned here (in the legend)."

→ We apologize for the mistake and have now corrected the figure legend accordingly.

6. "Figure 6e. In the high CAF infiltration, the CRIS classification seem to perform poorly. This should be better discussed."

→ We agree with the reviewer that the prognostic interaction between CAF infiltration and CRIS classification was discussed only tangentially in the original manuscript and we have now extended the presentation and the discussion of these data (Results, page 11). Indeed, despite the fact that both high CAF infiltration and CRIS-B assignment are significantly associated with poor prognosis in the whole population, CAF infiltrates are not prognostic within CRIS-B, and vice versa (Supplementary Figure 13). Albeit calling for confirmation in larger cohorts, this observation suggests that CRIS-B membership and high CAF infiltration identify rather alternative means to acquire analogous traits of cancer aggressiveness, whose negative prognostic impact is not further exacerbated by the combination of the two.

Reviewer #2:

"In the manuscript entitled "A global gene expression analysis of cancer-cell intrinsic transcriptional traits defines a new classification system for colorectal cancer with improved biological resolution and superior predictive and prognostic value" Isella and colleagues describe a new taxonomy for colorectal cancer (CRC) based on 5 newly discovered CRC intrinsic subtypes (CRIS). This taxonomy is obtained in patient derived xenografts (PDXs), hence it is based on cancer cells gene expression profiles (GEPs) that are not masked or "contaminated" by stromal contributions. The authors base their study on their previous observation that currently known CRC molecular subtypes are largely driven by GEPs from cancer associated fibroblasts (CAFs). The authors further characterize the newly discovered CRIS using several approaches, identifying biological themes associated with each class, and correlating these subtypes with distinct prognosis and response to cetuximab in public domain data.

The study is novel and leverages an extraordinary resource: a collection of over 500 transcription profiles derived from ~250 CRC patients. The release of such resource in the public domain as the result of the publication of this study will definitely benefit and positively impact the CRC research field. Furthermore, the newly discovered taxonomy might prove useful in understanding CRC biology, and in implementing strategies for improved clinical decision making. The enthusiasm for this study is diminished by the fact that the authors do not address the translational aspects of their discovery and do not fully develop their taxonomy into a tool for clinical use. It is not clear, indeed, how the CRIS classification will be used to assess patient prognosis or decide about treatment. The taxonomy is indeed based on more than 500 genes and the authors do not make any attempt to simplifying it to a manageable size and to implementing it as a clinical tool (for instance as it has been done with the breast cancer intrinsic subtypes and the PAM50 classifier). Another weakness is the fact that the prognostic and predictive value of this taxonomy is only investigated and assessed in two datasets despite the availability of many more cohorts from the public domain. Failing to do this substantially reduced the enthusiasm for and the significance of this study, making it unsuitable for publication on Nature Communications in its current status."

→ We thank the reviewer for these insightful general observations. We are convinced that fully addressing them, as detailed in the point-by-point replies below, has greatly increased the translational impact of the work. For the reviewer's convenience, all changes have been marked in red in a dedicated version of the manuscript (Isella et al_Manuscript_Marked Text). In a more general perspective, we have followed the reviewer's suggestion to investigate in greater detail the potential clinical implications of the CRIS taxonomy and we have built a simpler classification tool that is also more amenable to clinical portability, being composed of only 80 genes. To this aim, we have implemented a four-way strategy:

- 1- We have further validated CRIS-B prognostic significance by exploiting a new set of 1261 clinically annotated gene expression profiles assembled from 5 independent datasets of primary colorectal cancer (Results, page 12; Figure 6f; Supplementary Figure 17; Supplementary Table 11; see also reply to point 4c).
- 2- As an initial attempt to translate the CRIS taxonomy into a diagnostic tool amenable to clinical applications, we developed a single-sample classifier based on the top scoring pair algorithm (TSP) and its multiclass extension k-TSP. A TSP is a binary predictor based on the relative ranking of two measurements (e.g., the expression of a pair of transcripts), which switch order between two subclasses of samples. This approach can be extended to multiclass problems by identifying the TSPs associated with each pair-wise subclass comparison and then aggregating the votes across all gene pairs. We took advantage of such method to derive an algorithm for assignment of CRIS subtypes, which we named CRIS-TSP. To ensure cross-platform portability of the classifier, candidate TSP genes were challenged against a training dataset of 624 gene expression profiles from both PDXs and original tumors, obtained using multiple technological platforms (Supplementary Table 21). This process resulted in the selection of 40 gene pairs (Methods; Supplementary Table 22). When applied to the training set, CRIS-TSP demonstrated good concordance with the original CRIS classifier ($K = 0.7384$). CRIS-TSP was then applied to eight independent datasets, for a total of 2024 samples (Supplementary Table 21). When challenged against 6 gene expression datasets annotated for clinical outcome (total samples = 1487), CRIS-TSP assignments confirmed the poor prognosis of CRIS-B patients (log rank chi square, $P < 5 \times 10^{-5}$, HR= 1.6613, C.I. = 1.307 - 2.112; Supplementary Fig. 18, Supplementary Table 21). The classification concordance between CRIS and CRIS-TSP was, however, suboptimal in these datasets ($K = 0.6459$). We also found limited concordance when comparing CRIS-TSP classification of surgical resections of liver metastases versus their corresponding PDXs ($K = 0.4053$), overall suggesting that the performance of the CRIS-TSP classifier was less robust when applied outside the training set. To assess whether such reduced classification coherence was due to low classification

efficacy of the selected TSP genes, we reconstituted a NTP-based classifier using the same TSP transcripts (CRIS-NTP80, Supplementary Tables 21 and 22). When exploiting the CRIS-NTP80 algorithm for classification of the clinically annotated CRC datasets, the overall classification concordance with respect to the original full-size CRIS classifier was improved over CRIS-TSP ($K = 0.7149$). Consistently, the classification coherence between original tumors and PDXs was also increased ($K = 0.5577$) and similar to that obtained with the full-size NTP. CRIS-NTP80 also confirmed the prognostic significance of CRIS-B (log rank chi square, $P < 5 \times 10^{-4}$ HR= 1.581, C.I. = 1.246 - 2.006; Supplementary Fig. 18). Altogether, these data show that reducing the size of the CRIS classifier to 80 genes preserves most of its classifying capability across different gene expression platforms, and indicate the feasibility of deploying such a reduced gene set for single sample classification based on a TSP approach. These data also point to NTP-based CRIS categorization as the option of choice, whenever the goal is to classify already available gene expression datasets obtained by diverse technological platforms. Implementation of a clinically applicable TSP-based single-sample classifier is feasible for prospective classification of new samples, for which dedicated and standardized data-generation procedures can be adopted. These new observations have been included in the Results, pages 12-13, and Discussion, page 15.

- 3- To facilitate the fruition of the reduced classification tool by the community of researchers and clinicians, its algorithm has been now incorporated into an R package called CRISclassifier that is included in the manuscript as a supplementary file (Isella et al_CRISclassifier_R-package) and will be released on Bioconductor in case of manuscript approval. Through this tool, virtually any gene expression dataset can be classified by applying the exact same procedures that have been adopted in this study. To assist non-expert users, a document reporting step-by-step instructions to run the pipeline has been included in the package.
- 4- We revised the scheme presented in Figure 6g to clarify how we foresee the exploitation of the CRIS taxonomy for clinical use. To this aim, Figure 6g now depicts a decision tree that takes into account CRIS-B membership and the extent of CAF infiltration, together with the clinical indications, to refine the decisional process in the adjuvant setting.

REFERENCES

- Geman, D., et al. Classifying gene expression profiles from pairwise mRNA comparisons. *Stat Appl Genet Mol Biol* **3**, Article19 (2004).
- Tan, A.C., et al.. Simple decision rules for classifying human cancers from gene expression profiles. *Bioinformatics* **21**, 3896-904 (2005).
- Marchionni, et al. A simple and reproducible breast cancer prognostic test. *BMC Genomics* **14**, 336 (2013).
- Sung, J. et al. Multi-study integration of brain cancer transcriptomes reveals organ-level molecular signatures. *PLoS Comput Biol* **9**, e1003148 (2013).

“The methodology and the design used to perform this study are somewhat complex to follow. The authors used a multi-step procedure to 1) discover, 2) refine, and 3) validate a gene expression signature system associated with distinct molecular CRC subtypes. To achieve this, they combined unsupervised class discovery techniques with supervised class comparison and prediction methods. Throughout the study several parameters have been tuned, and these choices have not been completely justified. Similarly in some instances simpler and more straight forward approaches could have been used. The fact that this approach is published (it is basically the same one used by Sadanandam et al) does not make it less cumbersome.”

→ We appreciate the reviewer’s reasons for favoring a more straightforward approach to class discovery and class prediction. However, in our view it was important to apply exactly the same procedures previously adopted for human CRC samples (Sadanandam et al., 2013), so that differences in classification could not be ascribed to procedural divergences. Throughout the point-by-point reply below, this concept will be reiterated where relevant. Overall, we hope we will convince the reviewer that – coherently – when no obvious reasons emerged to favor other solutions, previously adopted criteria and thresholds were systematically preferred, and that this reinforces the significance of our findings.

REFERENCE:

- Sadanandam A. et al. A colorectal cancer classification system that associates cellular phenotype and responses to therapy. *Nat Med* **19**, 619-25 (2013).

“Nevertheless, findings are in general very well presented and figures are clear and consistent in using colors codes. A number of issues that should be addressed are discussed below.

In the study, although the analytical design is overly complicated, statistics is usually used appropriately. Generally each time a p-value is reported in the text he authors also reported the corresponding statistical test that was used to obtain it. The authors, however, should also assess whether there are deviations from normality and hence whether non parametric tests are more appropriate. Furthermore, in light of the recent statement from the American Statistical Association (<http://www.amstat.org/newsroom/pressreleases/P-ValueStatement.pdf>) it would be advisable to report effect sizes too. More comments on use of statistical tests is included in the detailed comments below.”

We have revisited statistics throughout the manuscript and adopted non parametric tests whenever deviations from normality were assessed. We have also reported effect sizes where appropriate.

Detailed comments:

1) *“Results Section: “CRC PDXs fail assignment to public transcriptional subtypes”.*

The authors fail to assign to public transcriptional subtypes CRC samples from their PDXs cohort. They speculate that this is due to the lack of stromal contribution. Although likely and plausible, more efforts should be devoted to prove this point. To support this hypothesis, indeed, the authors simply report the classification incongruence and the fact that PDXs are more similar to their matched tumors. Such findings prove that different results are obtained in PDXs and in tumors, and that matched pairs of PDXs and tumors are more similar to one another than random pairs. These results do not prove that this discrepancy is due to the absence of stromal contributions to GEPs.”

→ We agree with the reviewer that our results do not formally prove that the main cause of classification discrepancy between PDXs and their matched original counterparts is in fact attributable to the stromal components of the cancer transcriptome. However, we still hold that the data presented here strongly support the notion that stromal infiltration is the main cause of this discrepancy. The arguments in favor are:

- 1- Stromal transcripts are extremely enriched in the first principal component of original tumors (Supplementary Figure 1 and Supplementary Table 2) and are almost completely lost upon xenografting (Isella et al., 2015).
- 2- Stem-Mesenchymal classes heavily depend on stromal transcripts (Isella et al., 2015) and most of the discrepancies between PDXs and original tumors are observed for the Stem-Mesenchymal tumors classes (CCS $p < 0.0001$, CRCA $p < 0.005$, CMCS $p < 0.0014$ by Fisher’s exact test)
- 3- When comparing assignments to CRIS subtypes, which were obtained by exploiting the PDX profiles for class discovery, the classification concordance between PDXs and original tumors was greatly increased (tumor vs PDX kappa scores: 0.62 for CRIS; CCS = 0.256, CRCA= 0.28, and CCMS = 0.31 for the De Sousa e Melo, Sadanandam, and Marisa classifiers, respectively). This indicates that, instead of being ascribable to a generic and diffused alteration of the transcriptional profile, such inconsistencies are specifically caused by the loss of a component that drives classification in the original tumors, but loses classification efficacy in PDXs. Since we already know that the major transcriptional component that is lost upon tumor xenotransplantation in mice relates to the substitution of human stroma by murine infiltrates (see also point 1), it seems reasonable to infer that this host effect is also the major reason of misclassification.

In agreement with a point raised by Reviewer 1 (point 4), we appreciate that the best way to achieve formal validation of our interpretation would be to analyze transcriptional profiles obtained upon physical isolation of cancer cells (see also reply to Reviewer 1 - point 4). However, this approach would require a high number of samples for proper statistical considerations and class discovery purposes. This high-throughput format hardly reconciles with the procedures that are normally applied to separate cancer cells from the stroma – typically, cell sorting of dissociated tumors or laser-capture microdissection of histological slides – which are technically laborious and time-consuming. Honestly, we think that an effort of this kind goes beyond the scope of the present manuscript. At the same time, we acknowledge the relevance of the referee’s objection and, accordingly, have toned down some conclusions and commented on some limitations of the PDX setting in the Results (pages 4-5) and Discussion (page 15)

REFERENCE:

- Isella C. et al. Stromal contribution to the colorectal cancer transcriptome. *Nat Genet* **47**, 312-19 (2015).

Furthermore:

a) *"For the loss of classification rate in SSM classes the authors report p-values from chi square tests, however, it is not clear how chi-squared tests were used for this purpose. What is the null hypothesis tested?"*

→ We apologize for the lack of clarity and for omitting details on the null hypothesis tested. We have now included a declaration of the null hypothesis tested in the Results (page 4). Specifically, we assumed as null hypothesis that the fraction of cases classified as SSM in the PDX dataset is not different from the fraction of cases classified as SSM in the liver-metastatic human tumor samples.

b) *"For the difference in correlation between matched or random pairs the authors used t-tests and hence they should make sure that such correlations are normally distributed."*

→ We agree with the reviewer that t-test was probably not appropriate for this comparison. We have now substituted it with a Wilcoxon-Mann-Whitney test, which confirms significance of the results. Figure legends and results have been corrected accordingly.

2) *"Results Section: "Cancer-cell intrinsic transcriptional traits classify colorectal cancer".*

The procedure used to develop the CRIS taxonomy is not extensively explained and a flow-chart summarizing the procedures and steps involved would be beneficial. This is particularly important since the combination of unsupervised and supervised learning methods used is complex and somewhat cumbersome."

→ To comply with the reviewer's suggestion, a flowchart summarizing the steps followed to obtain the CRIS classifier has now been added to the Supplementary Figures (Supplementary Figure 20) and details have been included in the Methods section (page 19).

In addition to this:

a) *"The authors used 515 PDXs or so to learn the new taxonomy. However, there are only 225 patients and it is not clear how such redundancy was handled."*

→ As the reviewer rightly states, we worked on two or more PDX samples propagated from one original patient tumor for most cases. And, in most cases (61%), replicas were derived from independent tumorigraft lines and originated from spatially distinct areas of the same tumor of origin. Accordingly, for class discovery purposes, we deliberately decided to treat the redundant PDX samples as independent data points in order to take into account regional tumor heterogeneity when defining CRIS subtypes. We reasoned that the informative potential provided by treating such replicates as distinct entities would justify and override the potential biases introduced by bearing a certain degree of signal redundancy into the analysis. This choice is now better explicated in the Methods section (pages 16 and 18). Anyway, in the GEO gene expression dataset, the tumor ID and PDX line of origin are annotated for each PDX, so that further analyses are feasible.

<http://www.ncbi.nlm.nih.gov/geo/query/acc.cgi?token=wpetcoimndmnrngb&acc=GSE73255>

b) *"The authors have a matched set of tumor-PDXs pairs. It appears that they developed the taxonomy using all the PDXs and then compared results between PDXs and tumors in the matched subset. If so, the classification performance in tumors is somewhat "inflated", because matched cases are correlated. The authors could perhaps make a better use of their data by learning the taxonomy on the cases that are not matched, applying it then to the matched PDXs and tumors."*

→ The reviewer's interpretation is correct. Again, we appreciate the reviewer's criticisms but, similar to that discussed above, we remain convinced that the benefit of incorporating as much information as possible into the class discovery process outweighs lack of independent validation of the classification coherence between PDXs and original tumors. The primary goal of this study is to provide the most accurate cancer-cell-intrinsic transcriptional classification of CRC based on currently available data. Future studies – building on fully independent datasets of matched tumor-PDX profiles – will be required to precisely compare the coherence of CRIS classification versus that of other CRC transcriptional

classifiers. Here, we provide evidence that a PDX-derived classifier is more easily applicable to whole-tumor profiles than vice versa. This is robustly supported by CRIS classification of more than 3500 transcriptional profiles extracted from 16 independent datasets of primary and metastatic tumors. This said, we have been able to indirectly comply with the referee's suggestion when addressing the request – raised by both reviewers – to reduce the CRIS classifier to a format exploitable for clinical and diagnostic use. In particular, we prioritized 80 transcripts (see also above and answer to point 3 raised by reviewer 1) to derive both a single-sample classifier based on top scoring pairs (CRIS-TSP) and a reduced NTP classifier (CRIS-NTP-80). Such smaller set of genes was selected by using a training dataset of 624 gene expression profiles from both PDXs and original tumors, obtained using multiple technological platforms. In the case of PDXs, only those profiles lacking a matched original tumor were exploited in the training dataset ($n = 275$). Then, the performance of such reduced classifier was assessed in cases for which matched pairs of liver metastases and PDXs were available. We have obtained a concordance kappa score of 0.56 when comparing matched PDX-tumor pairs with CRIS-NTP-80. Altogether, these results strongly suggest that the core transcriptional traits sustaining CRIS subtypes are in their essence common between PDXs and their original counterparts, further supporting the key hypothesis of the study.

c) "The authors used non-negative matrix factorization (NMF) with 5 different initial conditions for the factorization algorithm and with k varying from 2 to 6. They identified an optimal partitioning into 5 clusters. Is this a real optimum, or other cluster numbers are similarly plausible (cophenetic coefficients are indeed really similar for $k=2$ and $k=5$)? How a different number of cluster would affect downstream analyses (e.g., association with functional processes, association with prognosis and response to therapy, and the likes)? In the end on TCGA data the 5 classes are clustered into two major groups... Or clustering, perhaps, is not an ideal tool..."

→ We agree with the reviewer that optimal partitioning is a delicate issue. We followed the same reasoning applied by Sadanandam et al., 2013: in case of similar clustering robustness (i.e., similar cophenetic coefficients), a higher subtyping resolution (i.e., a higher k) is preferable. This is the main reason why $k = 5$ was selected instead of $k = 2$ or $k = 3$. Moreover, partitioning samples in 5 clusters opens the opportunity to verify overlaps and differences with the 5 subtypes of Sadanandam and colleagues, obtained with the same procedures on human CRC samples.

$K = 5$ is in essence a more resolved partitioning of $k = 2$ and $k = 3$, as we now show in a caleydo view of CRIS hierarchical taxonomy (Supplementary Figure 19a). In this figure, $k = 2$ joins together CRIS-A and B and CRIS-C, D and E; $k = 3$ groups CRIS-A and B and CRIS-D and E, leaving alone CRIS-C. This hierarchy perfectly parallels the one obtained from the TCGA dataset (see Figure 3a) and the biological and functional characteristics of CRIS subtypes: indeed, CRIS-A and B are more inflammatory and MSI-like, while CRIS-C, D and E are characterized by higher WNT activity, with CRIS-C being specifically dominated by strong EGFR-dependent traits. Altogether, these data indicate that CRIS classes capture biologically relevant transcriptional states of colorectal cancer cells, and the extent of biological identity appears to be a function of the applied resolution. The rationale for this choice has been now included in the Methods (page 18).

REFERENCE:

- Sadanandam A. et al. A colorectal cancer classification system that associates cellular phenotype and responses to therapy. *Nat Med* **19**, 619-25 (2013).

d) "Furthermore, NMF was performed using the most variable genes ($SD=0.8$). What is the rationale for selecting an SD equal or larger than 0.8? How many genes were retained? Why not to use $SD=1.5$? Some rationale should be provided supporting this choice. As alternative the authors should show that this gene filtering procedure does not affect the number of clusters identified in downstream analyses."

→ Filtering the data for $SD \geq 0.8$ restricts the analysis to 1084 unique genes. Also in this case, this choice was made to parallel the procedures of Sadanandam and colleagues; We apologize for omitting these details, which have been now included in the Methods (page 18). We have also tested NMF with different SD filtering thresholds, and in all cases $k = 5$ maintained very high cophenetic correlation (Supplementary Fig. 19b).

REFERENCE:

- Sadanandam A. et al. A colorectal cancer classification system that associates cellular phenotype and responses to therapy. *Nat Med* **19**, 619-25 (2013).

e) *“The authors focused on samples best representing each subtype based on silhouette width removing samples with negative score values. How much the results would change using more strict (or relaxed) criteria for inclusion of the samples in subsequent analytical steps?”*

→ Again, we followed an established threshold, which has been widely adopted to select samples that belong to the core of each of the clusters for gene expression-based class discovery/clustering. Please find below a list of examples. In most of these cases (including ours), the selection of samples representative of each subtype through silhouette plots was aimed at minimizing the noise deriving from those samples whose membership to a specific subtype is ambiguous. With this goal in mind, the choice of a threshold of 0 seems the most appropriate. Indeed, as defined in the original paper presenting the silhouette plot method (Rousseeuw, 1987), negative scores are assigned to samples whose average distance to the other elements of the same cluster is higher than the average distance to the members of one of the other clusters. In other words, a sample with a negative silhouette score is on average more similar to the members of a cluster different from the one to which the sample has been assigned than to the other members of its class. Such samples are by definition ambiguous, as they have been assigned to a subgroup by the clustering algorithm but, based on average similarity, they should have been assigned to a different cluster. We now cite the original methodological paper by Rousseeuw in the Methods (page 18). At the same time, we feel that exploring in detail the possible downstream consequences of changing such threshold goes beyond the scope of this work.

REFERENCES:

- Rousseeuw PJ. Silhouettes: a graphical aid to the interpretation and validation of cluster analysis. *Journal of Computational and Applied Mathematics* **20**, 53-65 (1987).
- Sadanandam A. et al. A colorectal cancer classification system that associates cellular phenotype and responses to therapy. *Nat Med* **19**, 619-25 (2013).
- Verhaak R. et al. Integrated genomic analysis identifies clinically relevant subtypes of glioblastoma characterized by abnormalities in PDGFRA, IDH1, EGFR, and NF1. *Cancer Cell* **17**, 98-110 (2009).
- Zhang B. et al. Proteogenomic characterization of human colon and rectal cancer. *Nature* **513**, 382-7 (2014).
- Motomura K. et al. Immunohistochemical analysis-based proteomic subclassification of newly diagnosed glioblastoma. *Cancer Sci* **103**, 1871-79 (2012)
- Xu T. et al. Identifying Cancer Subtypes from miRNA-TF-mRNA Regulatory Networks and Expression Data. *PLoS One* **11**, e0152792 (2016).

f) *“The author used Significance Analysis of Microarrays (SAM) on 425 “good” samples, identifying 1083 genes differentially expressed across subtypes with FDR of 0.005. Similarly to what said before, how much the results would change using more strict (or relaxed) criteria for gene selection using SAM in subsequent analyses? Why did the authors used an FDR of 0.005 and not 0.1 or 0.001? How much choosing an alternative threshold would change downstream results?”*

→ In our experience with the previously defined CRC classifiers, the gene signatures of smaller size are those that preferentially lose performance when ported to independent datasets and different gene expression profiling technologies. Therefore, to allow for easier portability of the CRIS classifier to other technological platforms, we decided to opt for a final reference classifier containing 100-200 transcripts per subtype. The choice of 0.005 was based on the observation that it removed only one of the 1084 variable genes passing the SD threshold, while lowering the FDR threshold to 0.001 reduced the number of genes to 846, and we feared that this – along with subsequent implementation of additional filters –, would excessively restrict the size of the reference classifier.

g) *“Prediction Analysis for Microarrays (PAM) was applied to develop shrunken centroids for each class, further reducing the number of genes used to 903 genes, with an overall error rate in leave one out cross-validation of 0.035. Is this final set of 903 genes resulting from minimizing the cross-validation error in PAM? If so the authors should simply state it. Similarly to what observed above about NMF and SAM,*

also in this case the authors should prove that the parameters that they used did not affect downstream findings.”

→ The reviewer is correct: in accordance to guidelines, we applied PAM to minimize the global cross-validation error in leave-one-out analyses. This has been now more clearly specified in the Methods (page 18). The scope of including PAM in the pipeline was to outline the most robust CRIS core in the training set. On this basis, we selected the genes' configuration demonstrating lower error rate in cross-validation tests. We respectfully disagree with the reviewer's suggestion of testing other PAM thresholds. Indeed, the selection of a PAM configuration different from the one chosen (which, as mentioned above, was meant to provide global error minimization) would contradict the rational principle based on which PAM itself was applied. It is worth noting that of the 903 genes selected through PAM, only 801 were further considered for our analyses, because 102 were not scored positive for any specific class and consequently could not be exploited by NTP (see also below, reply to point h1).

h) *“To generate a cancer-cell intrinsic classifier not influenced by stroma-derived transcripts, the authors further selected 565 genes with unambiguous epithelial expression. This allowed to confidently classify 94% of the samples in the PDX training dataset. How these epithelial genes were identified? Why exactly 565? How did they “confidently” identified the 94% of the samples? Which gold-standard did the authors used? Isn't the PDXs set the training set? Shouldn't all samples from the training set be correctly classified?”*

i) *“In general, the gene filtering/selection procedure is not very well described and clarifications are need. In the methods section the authors state that “If a gene was positively associated to more than one class, it was assigned to the best scoring class only when the second highest value was at least 0.2 points lower. In all other cases, the gene was excluded from the analysis.” How many genes were filtered because of an ambiguous association with multiple clusters? How was the 0.2 threshold selected? How much this threshold affects the classification? Is there a reason to use only disjoint sets to classify patients?”*

j) *The authors also state that “the genes with an estimated stromal contribution above 50% were removed from the analysis.” How was this “stromal contribution” assessed? How is stromal contribution quantified? What is the rationale for using 50% as the filtering threshold? How this impact on downstream analyses?”*

→ We agree with the reviewer that some methodological aspects deserve better clarification. Most of the matters discussed below have been incorporated in the Methods section of the revised manuscript (pages 18-19), which has been extensively rephrased to allow easier understanding. A detailed point-by point reply follows here; the order of the questions has been changed to enable a more organic explanation:

j1) *“The authors also state that “the genes with an estimated stromal contribution above 50% were removed from the analysis.” How was this “stromal contribution” assessed? How these epithelial genes were identified? How is stromal contribution quantified?” [original points i and j]*

→ We exploited our previous data (Isella et al., 2015). In particular, the Supplementary Table 11 of this prior study reports the results of an analysis conducted on RNAseq data from CRC PDXs, in which for each gene the fraction of stromal (mouse) transcripts was calculated with respect to the total human + mouse transcripts. Through this, we estimated the stromal contribution to the overall transcript levels (stromal + epithelial) of each gene (see also the revised Methods, page 19).

REFERENCE:

- Isella C. et al. Stromal contribution to the colorectal cancer transcriptome. Nat Genet **47**, 312-19 (2015).

j2) *“What is the rationale for using 50% as the filtering threshold?” [original point j]*

→ When choosing 50% as threshold for maximal stromal contribution, we wanted to include in the classifier only those genes for which the majority of the signal (i.e. more than half of the reads calculated in Isella et al. 2015) was originated by epithelial cells. Our straightforward intention was to

exclude from the classifier those transcripts for which the signal contribution from stromal cells was expected to be higher than that from cancer cells. Through this selection we excluded from the classifier 84 genes (10.5%), while 717 features were further explored.

j3) "How this impact on downstream analyses?" [original point j]

→ To evaluate how filters affect the performance of the classifier, here and below we used the classification of 425 samples with positive scores in silhouette analyses obtained by applying the complete list of 801 genes selected by PAM as a reference. Following each step of the filtering procedure, NTP was run before and after the reduction of the classifier and the new classification was compared to the original one. In this case, as expected given the small number of removed genes (84), exclusion of the stromal genes had a negligible impact on PDX classification (1.7% discordance).

i1) "In general, the gene filtering/selection procedure is not very well described and clarifications are need. In the methods section the authors state that "If a gene was positively associated to more than one class, it was assigned to the best scoring class only when the second highest value was at least 0.2 points lower. In all other cases, the gene was excluded from the analysis." How many genes were filtered because of an ambiguous association with multiple clusters?" [original point i]

→ Starting from the 717 genes selected by SAM and PAM and filtered for low stromal contribution, this filter removed 152 genes.

i2) "Is there a reason to use only disjoint sets to classify patients?" [original point i]

→ The reason is primarily technical: NTP does not allow for redundancy across signatures, which requires the features identifying each subtype be unique. As a consequence, to deploy a NTP-compatible classifier we had to avoid any recurrence in the genes used to assign membership to the different CRIS subtypes.

i3) "How was the 0.2 threshold selected?" [original point i]

→ To this aim, we followed exactly the same procedure previously published in Isella et al., 2015, by which we generated NTP classifiers starting from the signatures of Sadanandam et al., 2013; Marisa et al., 2013; and De Sousa e Melo et al., 2013. Through this, we minimized procedural divergences between CRIS and the above classifiers, so that any difference in classification is most likely due to the different starting point: cancer-cell-specific vs mixed tumor/stroma transcriptomes.

REFERENCES:

- Isella C. et al. Stromal contribution to the colorectal cancer transcriptome. *Nat Genet* **47**, 312-19 (2015).
- Sadanandam A. et al. A colorectal cancer classification system that associates cellular phenotype and responses to therapy. *Nat Med* **19**, 619-25 (2013).
- Marisa L. et al. Gene expression classification of colon cancer into molecular subtypes: characterization, validation, and prognostic value. *PLoS Med* **10**, e1001453 (2013).
- De Sousa E Melo F. et al. Poor-prognosis colon cancer is defined by a molecularly distinct subtype and develops from serrated precursor lesions. *Nat Med* **19**, 614-18 (2013).

i4) "How much this threshold affects the classification?" [original point i]

→ By removing those genes that were not univocally associated to a specific subtype by PAM, we aimed at excluding features that would not contribute substantially to classification. This was confirmed by the evidence that this filtering (which removed 152 genes, see reply to point i1) introduced a classification discordance of only 1.5% from the previous step.

h1) "To generate a cancer-cell intrinsic classifier not influenced by stroma-derived transcripts, the authors further selected 565 genes with unambiguous epithelial expression. This allowed to confidently classify 94% of the samples in the PDX training dataset. How these epithelial genes were identified? Why exactly 565?" [original point h]

→ Here we summarize the filtering procedures: starting from the 903 genes selected by SAM and PAM, 102 genes did not have a positive PAM score for any of the classes (as a consequence of the centroid shrinkage procedure) and could not be used for NTP (see reply to point g). From the remaining 801 genes, 84 were removed because of stromal contribution > 50% (see reply to point h2), and further 152 genes were removed for ambiguous association (see replies to point i1 and i4). This leads to exactly 565 genes. All these filtering steps have been now summarized in Supplementary Figure 20.

h2) "How did they "confidently" identify the 94% of the samples? Which gold-standard did the authors used?" [original point h]

→ To obtain a confidence estimate, we exploited the fact that the NTP algorithm performs a Montecarlo analysis to estimate the false discovery rate of class assignment for each sample. According to the standard use of NTP (see Sadanandam et al., Nat. Med. 2013), we set the threshold for confident assignment at FDR < 0.2.

REFERENCE:

- Sadanandam A. et al. A colorectal cancer classification system that associates cellular phenotype and responses to therapy. Nat Med **19**, 619-25 (2013).

h3) "Isn't the PDXs set the training set? Shouldn't all samples from the training set be correctly classified?" [original point h]

→ As explained above, the "confidence" in classification was not based on agreement to a gold-standard reference. Instead, the robustness of sample membership to a specific subtype was estimated based on Montecarlo simulations. This represents one of the most significant added values of adopting NTP over other methods to assess subtype membership. The typical NMF class discovery approach forcedly assigns each sample to a cluster, and does not provide confidence metrics for the assignment of each sample. This is true not only for NMF, but also for other class discovery methods (see for example Verhaak R. et al. 2009, De Sousa E Melo et al. 2013). In such cases, samples that do not robustly belong to the discovered clusters are usually identified using silhouette width analysis (see also reply to point 2e). When using NTP, this task is intrinsically embedded into the class-assignment algorithm through Montecarlo-based FDR calculation. Again, holding on to analytical pipelines, criteria and thresholds previously adopted for the same task in independent datasets enables unbiased comparisons.

REFERENCE:

- Verhaak R. et al. Integrated genomic analysis identifies clinically relevant subtypes of glioblastoma characterized by abnormalities in PDGFRA, IDH1, EGFR, and NF1. Cancer Cell **17**, 98-110 (2009).
- De Sousa E Melo F. et al. Poor-prognosis colon cancer is defined by a molecularly distinct subtype and develops from serrated precursor lesions. Nat Med **19**, 614-18 (2013).

k) "At page 5 the authors state "All classes were well represented in both the PDX and CRC-LM datasets, displayed similar distributions, and were sustained by similar underlying transcriptional traits". It is not clear what this really means, the authors should elaborate more on this."

→ To improve the clarity of this sentence, we have modified the text as follows: "All CRIS subtypes maintained a similar fraction of assigned cases in the PDX and CRC-LM datasets (Figure 2d). Submap analysis (Hoshida et al.) confirmed that each subtype was associated with similar underlying transcriptional traits in the PDX and CRC-LM datasets (Supplementary Fig. 3)" (Results, page 5).

REFERENCE:

- Hoshida Y. et al. Subclass mapping: identifying common subtypes in independent disease data sets. PLoS One 2, e1195 (2007).

j) *"The authors tested their CRIS classifier on two independent gene expression datasets of primary CRCs (TCGA and GSE14333). Why did they use only these 2 datasets (there are many more available)?"*

→ Also to comply with a request from Reviewer 1, we have now extended CRIS classification to a total of 3738 CRC samples extracted from 16 independent datasets; of these, 3396 (90.8%) were confidently classified by CRIS (FDR < 0.2) through NTP. The gene expression profiles of the additional datasets examined were obtained using different technologies, including Affymetrix, Illumina RNAseq, and Illumina BeadArrays. This further indicates that the CRIS classifier can be successfully applied across different technologies and experimental settings. The data are now presented in the Results (page 6), Figure 2i, and Supplementary Table 11

3) *"Results Section: "Major peculiarities of CRIS classes"."*

a) *"The authors showed that MSI and MSS samples are not equally partitioned across CRIS subfamilies in the TCGA dataset. Is this difference significant as it appears from the figure (especially between CRIS-A and the rest of the CRIS groups)?"*

→ MSI cases are significantly enriched in CRIS-A vs all other cases, as evaluated by Fisher's exact test ($p = 1.73 \times 10^{-7}$). However, after removing CRIS-A samples from the dataset, CRIS-B was also significantly enriched in MSI cases. For this reason, we now present the Fisher's p-values of A-B vs C-D-E: "Specifically, MSI tumors were predominantly assigned to CRIS-A and – to a lower extent – to CRIS-B (Fisher's exact test, CRIS-A/B against all other samples, $P < 5 \times 10^{-10}$; Figure 3d; Supplementary Tables 1 and 11) (Results, page 6)."

b) *"The authors stated that "These two classes proved to be also enriched for right-colon tumors featuring mucinous histology, a hyper-mutator genotype, and a CpG island methylator phenotype". From what this reviewer can understand from Figure 3D the enrichment of samples with these characteristics is in CRIS-A compared to CRIS-B, CRIS-D, and CRIS-E, not compared to CRIS-C. A chi-squared test could be used to assess whether these differences are significant or not."*

→ Indeed, Figure 3d shows quite clearly an enrichment in the fraction of red dots (positive cases) for both CRIS-A and B, while the fraction of positive cases for CRIS-C is similar to that of subtypes D and E. Again we have now aggregated CRIS-A and B in the Fisher's test analysis, and modified the text in the Results as follows: "These two classes proved to be also enriched for right-colon tumors featuring mucinous histology, CpG island methylator phenotype (CIMP) and a hypermutator genotype (Fisher's exact test, CRIS-A/B against all other samples, $P < 5 \times 10^{-3}$, $P < 5 \times 10^{-10}$, $P < 1 \times 10^{-11}$ and $P < 5 \times 10^{-7}$ respectively; Fig. 3d). Of note, some of these MSI-associated features were also shared by the MSS members of CRIS-A ("MSI-like" samples; Fisher's exact test, CRIS-A MSS samples against all other MSS samples, $P < 5 \times 10^{-6}$ and $P < 0.05$ for mucinous histology and CIMP phenotype, respectively; Figure 3d)." (Results, page 6)

c) *"The authors described several copy number alterations associated with CRIS in TCGA data - e.g., gains of entire chromosome arms in CRIS-C (Chr8p, Chr20p and Chr20q) and CRIS-E (Chr13q), focal amplification of 8q.24.21 in CRIS-C, etc... From Figure 4a, Chr13q clearly appears to be enriched also in CRIS-C, and perhaps also in CRIS-D. Furthermore additional red "bands" are evident in Figure 4a in other classes, hence it is not clear what is the rationale to focus on the selected regions. How was this enrichment defined? Did the authors set a cut-off to define enrichment, then did they count the enrichment events in a given genomic region, and then finally compare proportions across CRIS classes? Or is this analysis simply exploratory?"*

→ We apologize for the lack of clarity in formalizing the criteria used to select the copy number changes highlighted in Figure 4. Indeed, after reviewing the whole process, we have come to acknowledge that

deploying different thresholds for the categorization of broad and focal alterations, as we did in the original version of the manuscript, can be perceived as confusing and somehow arbitrary (see also points d and f). At the same time, in practical terms SNPchip-derived copy number data are complex and noisy, which renders the selection of a unique threshold for proper categorization of both broad and focal changes difficult. Thus, we have decided to simplify our approach. On one side, we have removed the highlights in Figure 4a and have cited in the Results only some of the broad copy number alterations for which highest average variations were observed in individual subclasses, with a merely descriptive intent. We concur with the reviewer that this has to be intended as a pure exploratory analysis deserving further consolidation in future studies; accordingly, we explicitly state this in the Results section (page 6). On the other side, taking advantage of the official categorization of focal copy number changes available for most samples of the TCGA dataset (and reported in Supplementary Table S1), we have formalized the results relative to Figure 4b; in particular, focusing on copy number variations with potential functional relevance we have highlighted only those significantly enriched in one CRIS subtype when compared to the other MSS samples by Fisher's exact test. Results have been modified accordingly (page 7).

d) *"In the methods section the authors state: "For each sample, all the regions with an absolute segmented value greater than 0.3 were categorized as altered." Is there any rationale for selecting the 0.3 threshold?"*

→ The 0.3 threshold is the standard, more conservative, threshold for calling a copy number alteration for a segment in GISTIC analysis of a single sample (Zack et al). This is now mentioned in the Methods (page 17).

REFERENCE:

- Zack T. et al. Pan-cancer patterns of somatic copy number alteration. *Nat Genet* **45**, 1134-40 (2013).

e) *"Furthermore they state: "The copy number load was calculated as the number of nucleotides included in such altered regions, relative to the sum of all the nucleotides covered by the sample segments." The authors should elaborate more about such "copy number load". Indeed, while it is clear what the authors mean by "number of nucleotides in the altered regions", it is not clear what they mean by "sum of the nucleotides covered by the sample segments". All the segments identified in the genome of the patient under considerations? The segments in the region for that patient? Clarifications are needed here."*

→ We have now rephrased the statement to make it more clear: "The copy number load was calculated as the number of nucleotides included in such altered regions, relative to the sum of all nucleotides in all the segments identified in the genome of the patient under consideration" (Methods, page 17).

f) *"Always in the method section the authors state: "All the regions scoring outside of the noise thresholds defined by GISTIC (0.1 and -0.1) were categorized as altered." How selecting this or another threshold might affect the results?"*

→ As a consequence of the simplified analytical pipeline described in point c, this threshold was not exploited anymore.

g) *"The authors compared mutation prevalence in CRIS classes and report Fisher's exact test p-values. How did the authors perform the Fisher's exact tests? Did they compare one CRIS class to the remaining ones lumped together? This applies for BRAF, KRAS, and TP53."*

→ The reviewer's interpretation is correct. Each CRIS class was compared with all the others lumped together, including unassigned samples. We apologize for the lack of clarity and have now included this information in the Results (page 7).

h) "Furthermore, how did the authors test for enrichment and depletion (as described for TP53)? Did they perform each pair-wise comparison between the CRIS classes? If so, did they correct for multiple testing?"

→ As explained above in reply to point g, enrichment was not tested by pair-wise comparison but by Fisher's exact test. Regarding multiple testing correction, everywhere in the manuscript we present the raw p values. When multiple testing correction is required, we have applied the Benjamini-Hochberg procedure and considered results significant when the FDR was below 0.2. We now specify the details of each enrichment test in the Results (page 7) and define the statistical analysis used for multiple testing in the Methods (page 23).

i) "To functional characterize the CRIS classes the authors use a combination of approaches based on GSEA. At page 7 they state: "we ranked samples by scores calculated from manually curated signatures relevant for CRC biology, and employed the GSEA algorithm to test each CRIS subtype for enrichment in high-rank samples." This statement is really confusing. Which manually curated genes sets were used and how were they selected? How were the high rank samples selected? What does this enrichment ultimately represent?"

→ We agree that the description of the Sample Set Enrichment Analysis (SSEA) needed improvement, and have modified the text in the Results and the Methods accordingly (pages 7-8 and 22, respectively). For the reviewer's convenience, we answer below to the individual questions.

i1) "Which manually curated genes sets were used and how were they selected? "

→ The ten gene sets included in the SSEA analysis are reported in Supplementary Table 14, together with the respective references. In principle, any possible gene set could have been used for this analysis. However, the aim was to concentrate on phenotypic traits that had already been associated with CRC, and verify whether such traits were asymmetrically distributed across the CRIS subtypes. Therefore, the list includes gene sets (signatures) specifically developed in CRC and thus associated with known CRC molecular features. More in detail:

- 1- "Bottom vs Top crypt": this bottom crypt signature was selected as representing different stages of differentiation occurring during colonic epithelial cell renewal, proliferation, and differentiation (Kosinski et al., 2007).
- 2- "EGFR superfamily": a list of ligands associated with EGFR pathway activation, as reported in Zanella et al., 2015.
- 3- "LGR5-positive colon cells": a signature associated with positivity to LGR5 expression, a known marker of intestinal stem cells (ISCs) in colonic epithelium (Merlos-Suarez et al., 2011).
- 4- "Epithelial to mesenchymal transition": This signature is associated with epithelial-mesenchymal transition in CRC tumors (Loboda et al., 2011).
- 5- "MSI vs MSS": a signature associated with microsatellite instability (Banerjea et al., 2004).
- 6- "Mucinous vs non Mucinous Adenocarcinoma": a signature discriminating between the two major histological subgroups of CRC (Melis et al., 2010).
- 7- "Neuroendocrine markers": genes associated with neuroendocrine differentiation of intestinal cells (Hofsli et al., 2008).
- 8- "Paneth cells markers": a signature distinguishing Paneth cells from other intestinal cells (Wang et al., 2011).
- 9- "Stem": a signature associated with cell positivity for EphB2, an alternative ISC marker (Merlos-Suarez et al., 2011).
- 10- "WNT signaling activity": these genes were extracted from two databases, in which direct transcriptional targets of the WNT pathway in CRC are defined (http://web.stanford.edu/group/nusselab/cgi-bin/wnt/target_genes and Herbst et al., 2014).

REFERENCES:

- Kosinski C et al. Gene expression patterns of human colon tops and basal crypts and BMP antagonists as intestinal stem cell niche factors. Proc Natl Acad Sci USA **104**, 15418-15423 (2007).
- Zanella et al. IGF2 is an actionable target that identifies a distinct subpopulation of colorectal cancer patients with marginal response to anti-EGFR therapies. Sci Transl Med **7**, 272ra12 (2015).

- Merlos-Suarez A. et al. The Intestinal Stem Cell Signature Identifies Colorectal Cancer Stem Cells and Predicts Disease Relapse. *Cell Stem Cell* **8**, 511-524 (2011).
- Loboda A. et al. EMT is the dominant program in human colon cancer. *BMC Med Genomics* **4**, 9 (2011).
- Banerjea, A. et al. Colorectal cancers with microsatellite instability display mRNA expression signatures characteristic of increased immunogenicity. *Mol. Cancer* **3**, 21 (2004).
- Melis, M. et al. Gene expression profiling of colorectal mucinous adenocarcinomas. *Dis Colon Rectum*, **53**, 936-943 (2010).
- Hofsl E. et al. Identification of novel neuroendocrine-specific tumour genes. *Br J Cancer*. **99**, 1330-1339 (2008).
- Wang et al. Paneth cell marker expression in intestinal villi and colon crypts characterizes dietary induced risk for mouse sporadic intestinal cancer. *Proc Natl Acad Sci U S A*. **108**, 10272-10277 (2011).
- Herbst A. et al. Comprehensive analysis of β -catenin target genes in colorectal carcinoma cell lines with deregulated Wnt/ β -catenin signalling *BMC Genomics* **15**, 74 (2014).

i2) *"How were the high rank samples selected? What does this enrichment ultimately represent?"*

→ For a given gene set, a transcriptional score was calculated for each sample by subtracting the average expression of the genes negatively associated with the phenotype from the average expression of the genes positively associated with the phenotype. In this way, the resulting score defines to what extent the sample displays the phenotype captured by the signature. Then, all samples were ranked by the score, and enrichment for high-rank samples was estimated for each CRIS subtype ("sample set") using the same statistics as GSEA. This is the reason why we named this approach SSEA.

j) *"The authors further "estimated the enrichment of each subtype in a number of mitogenic/anti-apoptotic autocrine loops by ranking the samples based on concomitant ligand-receptor expression." This is an interesting approach. The authors might want to consider the manuscript by Boca et al, in which samples-centered enrichment was introduced for the first time () as an example of the procedures one should put in place and use to assess the reliability and robustness of a new test/approach."*

→ We thank the reviewer for the suggestion, and have checked the manuscript by Boca and colleagues. In that work, no patient ranking is deployed to verify preferential high- or low-rank for a given patient subpopulation. As explained in the reply to point i2, SSEA exploits the algorithm and implementation of GSEA, including permutation-based estimation of FDR and Enrichment Score. Therefore, SSEA should not be considered a new test but rather a GSEA in which patients are ranked instead of genes. These specifications are now reported in the Methods (page 22).

k) *"Enrichment analyses can only show what it is represented by the functional classes (i.e., the gene sets) selected and fed in the enrichment test itself. How much bias is introduced by this approach in selecting "relevant CRC gene sets"? Why the authors did not use an unbiased approach based on a larger and comprehensive collection of gene sets (for instance as available from MSigDB)? Finally, how did the authors handle multiple testing resulting from the use of multiple gene sets (if they actually performed a test to identify the most strongly associated processes)?"*

→ Indeed, we carried out unbiased systematic functional enrichment analysis at the gene-set level by GSEA (which includes multiple testing correction) using MSigDB – specifically, using the "Hallmark" gene sets – with the complete results reported in Supplementary Table 13. While GSEA was an unbiased analysis, SSEA was a supervised, complementary approach with a focus on specific CRC features, as explained in the reply to point i1. We now better specify the complementarity of these approaches in the Results (pages 7-8).

l) *"Based on the radar plots the authors state that " Both enrichment analysis and phenotypic signatures were concordant in attributing secretory and MSI-like features to CRIS-A, in agreement with the mucinous histology of this subgroup detected in the TCGA dataset (Fig.4E)." Figure 4E is, indeed, very effective in showing enrichment across different functional gene sets for each CRIS class identified, however the*

authors should investigate the relationships among the different gene lists used to make sure that such enrichment is driven by different genes and not by a common core set of genes shared by the different lists used for the analysis."

→ We have checked the intersections of the various signatures and found that the great majority of the genes (92%, 1212/1315) was included in only one signature. In other words, less than 10% of the genes were included in more than one signature. Moreover, only 8 of the 108 "redundant" genes were present in more than two signatures. Thus, it is very unlikely that common enrichments for these signatures are due to shared sets of genes.

m) "The authors reported that "blinded pathological inspection of images from TCGA samples revealed that CRIS-B members included a large number of poorly differentiated tumors, in which the glandular architecture of the original tissue was completely lost or barely detectable." Is this association anecdotal or there is a significant enrichment of undifferentiated tumors in the CRIS-B group?"

→ The finding is preliminary; indeed, we did not mention any statistical evaluation. Accordingly, we have now added the sentence: "this preliminary observation deserves future, more extensive exploration" (Results, page 8).

n) "In general, are the different functional and phenotypic characteristics associated with the CRIS classes quantifiable in some way? Did the author quantify somehow all these associations? Are the gene sets represented in Figure 4E and 4F all those that were investigated or only those enriched among all the investigated ones? In the figure legend a p-value from a Fisher's exact test is mentioned, however it is not really clear where and how this test was used."

→ The tests used for panels 4a-d are now precisely mentioned in the Results (page 7). As explained above, all associations displayed in Figure 4e and 4f are significant based on GSEA statistics. The NES value quantitatively reports the association as calculated by GSEA.

The complete results of the enrichment analyses, i.e. the GSEA reports, are provided in Supplementary Tables 13 (GSEA – MsigDB-hallmarks), 14 (SSEA for CRC gene sets) and 15 (SSEA for autocrine loops). Among the significant associations reported in Supplementary Tables 13-15, Figures 4e and 4f report those that we considered biologically and clinically more informative.

The sentence mentioning the Fisher's exact test was erroneously placed in the legend to Figure 4f. We have now moved it to its correct position, in the legend to panel 4d. We apologize for this oversight.

o) "The authors show that there is limited overlap between sample partition based on CRIS and CMS. Can the authors test for the lack of the association? Furthermore, Figure 4G is mapping samples between CMS subtypes and CRIS groups, however, cross-tabulation would probably show this relationship more effectively and in a quantitative way."

→ We thank the reviewer for the observation, and have included cross-tabulation of CMS and CRIS (using the TCGA dataset) in Supplementary Table 16. To evaluate the association between the two classification criteria we have performed pairwise comparisons of CRIS and CMS classes followed by Fisher's exact test. The results are included in Supplementary Table 16. Odds ratio and Fisher's test statistics provided formal substantiation to the associations between CMS and CRIS originally described in the manuscript. Moreover, a weaker but significant association between CRIS-B and CMS4 was detected, which has been now included in the Results (page 9). Overall, the detected associations were never bi-univocal, confirming our conclusion that sample partitioning by the two classifiers displays only "limited overlap".

p) "At page 8 the authors state: "As shown by the heatmaps in Fig. 4G, asymmetric distribution of stromal scores was clearly detected in the CMS partition but not in the CRIS partition". The figure clearly shows that CAFs genes are spread over different CRIS groups, while these are basically all grouped in CMS4. It is not clear, however, what the author means by "stromal score" here... Perhaps this lack of clarity is due to space constraints, but an effort should be made to clarify what these heatmaps exactly show."

→ The stromal scores were previously calculated for TCGA samples in Isella et al., 2015. In particular, we obtained such scores from the Supplementary Table 17 of that work. In this study, each stromal score (Cancer associated fibroblast/ CAF-score, Leucocyte-score and Endothelial-score) was calculated by averaging the expression of the genes of the respective signature, reported in Supplementary Table 14 of Isella et al., 2015. This is now more clearly mentioned in the Results (page 9).

REFERENCE:

- Isella et al. Stromal contribution to the colorectal cancer transcriptome. *Nat Genet* **47**, 312-9 (2015).

4) "Results Section: "CRIS predicts response to anti-EGFR antibodies"

The authors used the GSE14333 dataset to study the association of CRIS classes to anti-EGFR antibodies response. They have shown that CRIS-C subtype is over-represented in the cetuximab-sensitive patients, and that the opposite happened for CRIS-A. These associations were investigated using cross-tabulation and Fisher's exact tests, reporting odds ratio and p-values. This association persisted also after removing from the analysis the cases harboring genetic alterations conferring resistance. The authors also investigated a set of genes indicative of EGFR pathway activity, which showed higher expression in CRIS-C tumors."

→ For the sake of accuracy, this analysis was conducted using the GSE5851 dataset, a set of liver metastatic CRC annotated for response to cetuximab.

a) *"The authors should specify to which group CRIS-C patients were compared by t-test when they evaluated EGFR pathway activity. The other classes lumped together? Furthermore, is the use of the t-test appropriate (normally distributed data)?"*

→ To avoid the normality issue, we have repeated the analysis using the Wilcoxon rank sum test. This new analysis did not substantially alter the resulting p-values.

The EGFR score has been compared between the following groups:

- CRIS-C vs all other samples, including unassigned cases ($p = 9.537 \cdot 10^{-9}$)
- Cetuximab-sensitive vs resistant cases, in samples wild-type for all the genetic markers of resistance identified so far ($p = 8.294 \cdot 10^{-5}$)
- Cetuximab-sensitive vs resistant cases, in non-CRIS-C wild-type for all the genetic markers of resistance identified so far ($p = 0.0001051$)

We have now modified the Results (page 10) to include such changes.

b) *"The authors used multiple linear regressions and demonstrated that genetic markers, CRIS classes, and EGFR pathway activity have independent predictive value. It is not clear, however, how the gene expression profile for EGFR pathway activity was summarized for each patient in order to fit the regression. Furthermore, the authors did not clarify which is the response variable used in such regression model. If the response to treatment is the outcome, how was this measured? Is the response sensitivity to treatment? If so, isn't the outcome binary and a logistic model should be used instead of a linear regression?"*

→ The EGFR pathway activity score was calculated by averaging the expression of the genes composing the gene set defined by Zanella et al., 2015.

The response to cetuximab is indeed a binary outcome; therefore, we have applied a logistic regression model as suggested by the reviewer. This alternative analysis, which now substitutes the previous one in Supplementary Table 18, confirms the previous results and is now presented in the Results, page 10.

REFERENCE:

- Zanella et al. IGF2 is an actionable target that identifies a distinct subpopulation of colorectal cancer patients with marginal response to anti-EGFR therapies. *Sci Transl Med* **7**, 272ra12 (2015).

c) *"To explore CRIS prognostic impact the authors "examined the association of CRIS subtypes with disease-free survival (DFS) of 290 patients included in a publicly available and clinically annotated CRC dataset (GSE14333)". Aren't there more datasets with survival information available? More effort should*

be put into this analysis to strengthen these prognostic associations. Aren't survival data also available for TCGA?"

→ As stated above (see reply to Reviewer #1, point 1), in the original manuscript we evaluated the performance of CRIS on approximately 700 gene expression profiles extracted from 3 independent CRC datasets from both primary tumors and liver metastases. Now, we have extended the analysis to a much larger collection of samples. Collectively, CRIS has been newly validated on a total of 3738 CRC samples obtained from 16 independent datasets. This has allowed us to confirm CRIS prognostic implications in an independent cohort of 1261 samples, which was assembled by combining data from 5 independent datasets (including the TCGA). Also in this case, both CRIS-B and high CAF content predicted poor prognosis (Supplementary Figure 15), but the combination of both outperformed either of the individual indicators (log rank chi square, $P < 1 \times 10^{-7}$; Fig. 6f). These new results have been added to the manuscript (page 12).

d) "At page 10 the authors state: "When challenged in tumors with high or low stromal content, as estimated by a transcriptional CAF score, CRIS-B was informative specifically in low-CAF tumors (Fig. 6D and E; log rank chi square, $P < 5 \times 10^{-5}$). These results prompted us to test a prognostic indicator that integrates CAF-score and CRIS-B assignment in an "or-based" algorithm. The combined classifier performed better than either CAF score or CRIS-B alone". How does this combined classifier work? No details about the classification rule employed were found in the manuscript."

→ We agree that the definition of the "or-based" classification rule was not clear. We have now rephrased the text in the results section (page 11) stating that "all patients categorized as CRIS-B or high-CAF, or assigned to both groups, were assumed to have a poor prognosis".

e) "The authors further state: "the integrative deployment of CRIS classes and CAF score was able to discriminate a relatively large subset (30%) of poor-prognosis patients who relapsed in 5 years in more than 40% of cases (sensitivity of 0.68 and specificity of 0.75 for 5-year DFS)". How about comparing classifications results with CRIS, CAF, and the combined classifiers using AUC? How about using more than just one dataset?"

→ AUC analysis is not applicable to our case, as assignment to CRIS-B is by definition a categorical variable. The suggestion of extending the combined prognostic analysis to other CRC datasets has been addressed (see reply to point c above).

5) "On reproducible research. The importance of reproducible research and data sharing has been strongly remarked by multiple Institutions and Publishers (see for instance the US Institute of Medicine report at <http://www.nature.com/news/lapses-in-oversight-compromise-omics-results-1.10298>). For this reason, and in accordance to recommendations about reproducible research (see "Reproducible research in computational science." by Peng, R.D. in Science 334, 1226-1227, 2012), all the scripts and code used for performing the analyses should be provided as supplementary material. To this end both the PDXs and tumor datasets used in this study (GSE76402 and GSE73255) are not currently available and hence none of the analyses performed could be reproduced and evaluated by this reviewer. For a true assessment of the validity of the findings presented in this study a link to these datasets should have been provided to the reviewers."

→ We provided the link to GSE76402 and GSE73255 at the time of initial manuscript submission. It is unfortunate that the reviewer was unable to access the datasets. We hope the link will be now made available for peer review.

REVIEWERS' COMMENTS:

Reviewer #1 (Remarks to the Author):

I am satisfied and impressed by the authors replies and improvements in the revised version of the manuscript, and have no hesitation to recommend it for publication

Reviewer #3 (Remarks to the Author):

General comments

The authors provided mostly satisfactory revisions and clarifications in response to the criticisms. One major issue, further development/refinement of the signature for potential clinical translation, was addressed by reducing the number of genes to 80 and adopting a new algorithm to optimize its performance. In addition, several technical details and justification for the choices of methods and related parameters were provided. Several minor issues are listed below.

Specific comments

- (1) It would be worth briefly mentioning the regional tumor heterogeneity possibly recovered by multiple PDXs derived from the same patients in the Results section.
- (2) It would be more ideal and simpler to stick to Fisher's exact test throughout the paper instead of using chi-square test.
- (3) "Wilcoxon Mann Whitney test", "Wilcoxon rank sum test": Should be unified to either "Mann-Whitney U test" or "Wilcoxon rank-sum test".
- (4) Is the rationale for the 50% filtering threshold (reviewer #2, comment 2j2) mentioned in the text?

Point-by-point reply to reviewers' comments:

Reviewer #1:

"I am satisfied and impressed by the authors replies and improvements in the revised version of the manuscript, and have no hesitation to recommend it for publication"

→ We thank the reviewer for his/her positive comments.

Reviewer #3:

"The authors provided mostly satisfactory revisions and clarifications in response to the criticisms. One major issue, further development/refinement of the signature for potential clinical translation, was addressed by reducing the number of genes to 80 and adopting a new algorithm to optimize its performance. In addition, several technical details and justification for the choices of methods and related parameters were provided. Several minor issues are listed below."

→ We thank the reviewer for his/her positive evaluation and fruitful suggestions, which have been addressed as detailed below.

"It would be worth briefly mentioning the regional tumor heterogeneity possibly recovered by multiple PDXs derived from the same patients in the Results section."

→ We have followed the reviewer's suggestion and included an explicit reference to the heterogeneity of the original tumors in the Results section, page 4.

"It would be more ideal and simpler to stick to Fisher's exact test throughout the paper instead of using chi-square test."

→ Following a careful revision of all the statistical tests applied in the study, we would opt to maintain our original choice for the tests used. In particular, throughout the manuscript we adopted coherent rules to guide our selection:

- a. We used Fisher's exact tests whenever a 2x2 contingency table was applied to a cohort
- b. We adopted log rank chi-square tests specifically for survival analyses
- c. We exploited chi-square tests when comparing the prevalence of subgroups in different cohorts

"'Wilcoxon Mann Whitney test', 'Wilcoxon rank sum test': Should be unified to either 'Mann-Whitney U test' or 'Wilcoxon rank-sum test'."

→ We apologize for these discrepancies, which were due to an incorrect definition of the tests used in the replies to reviewer 2 (point 1, page 7 of the letter in response to reviewers' comments). We have reviewed the whole manuscript and double-checked for consistency throughout the text.

"Is the rationale for the 50% filtering threshold (reviewer #2, comment 2j2) mentioned in the text?"

→ We have followed the reviewer's suggestion and included such rationale in the Results section (page 5).